# *In silico* co-factor balance estimation using constraint-based modelling informs metabolic engineering in *Escherichia coli*

**Laura de Arroyo Garcia, Patrik R. Jones** *

Department of Life Sciences, Imperial College London, London, United Kingdom

* p.jones@imperial.ac.uk

**Data Availability Statement:** All relevant data are within the manuscript and its Supporting Information files.

## Abstract

In the growing field of metabolic engineering, where cells are treated as 'factories' that synthesize industrial compounds, it is essential to consider the ability of the cells' native metabolism to accommodate the demands of synthetic pathways, as these pathways will alter the homeostasis of cellular energy and electron metabolism. From the breakdown of substrate, microorganisms activate and reduce key co-factors such as ATP and NAD(P)H, which subsequently need to be hydrolysed and oxidized, respectively, in order to restore cellular balance. A balanced supply and consumption of such co-factors, here termed *co-factor balance*, will influence biotechnological performance. To aid the strain selection and design process, we used stoichiometric modelling (FBA, pFBA, FVA and MOMA) and the *Escherichia coli* (*E.coli*) core stoichiometric model to investigate the network-wide effect of butanol and butanol precursor production pathways differing in energy and electron demand on product yield. An FBA-based co-factor balance assessment (CBA) algorithm was developed to track and categorise how ATP and NAD(P)H pools are affected in the presence of a new pathway. CBA was compared to the balance calculations proposed by Dugar et al. (Nature Biotechnol. 29 (12), 1074–1078). Predicted solutions were compromised by excessively underdetermined systems, displaying greater flexibility in the range of reaction fluxes than experimentally measured by $^{13}$C-metabolic flux analysis (MFA) and the appearance of unrealistic futile co-factor cycles. With the assumption that futile cycles are tightly regulated in reality, the FBA models were manually constrained in a step-wise manner. Solutions with minimal futile cycling diverted surplus energy and electrons towards biomass formation. As an alternative, the use of loopless FBA or constraining the models with measured flux ranges were tried but did not prevent futile co-factor cycles. The results highlight the need to account for co-factor imbalance and confirm that better-balanced pathways with minimal diversion of surplus towards biomass formation present the highest theoretical yield. The analysis also suggests that ATP and NAD(P)H balancing cannot be assessed in isolation from each other, or even from the balance of additional co-factors such as AMP and ADP. We conclude that, through revealing the source of co-factor imbalance CBA can facilitate pathway and host selection when designing new biocatalysts for implementation by metabolic engineering.

**Funding:** This work was supported by the Biotechnology and Biological Sciences Research Council, London, UK, DTP grant BB/J014575/1 - BB/M011178/1 to L.A.G. The funders had no role in the study design, data collection and analysis, decision to publish, or preparation of the manuscript.

**Competing interests:** The authors have declared that no competing interests exist.

## Author summary

The chemicals industry is a major contributor to greenhouse gas emissions and desperately requires more sustainable alternatives. Genetically engineered microorganisms can be used as 'bio-factories' to manufacture chemicals, replacing those currently sourced from fossil fuels or unsustainable tropical plant agriculture. However, due to the complexity of biology, the features that render one bio-factory design more efficient than others are difficult to identify. Computational modelling of such designs can enable the selection of optimally performing designs, but it remains challenging as biology is complex and not fully understood. Microorganisms require energy for their own growth and maintenance, but also to convert molecules into desired target products. The supply and consumption of such energy is through co-factors, and the balance of such co-factors influences the performance of the engineered bio-factories. This study developed a computer-aided approach for quantification of the co-factor balance of bio-factories. Using the chemical n-butanol as a case study, our study explores the impact of variant bio-factory designs with differing co-factor balance on the potential efficiency of biomanufacturing. We provide insights into the relative balance of different designs and provide a computational framework to select the best-performing designs.

## Introduction

Metabolic engineering, also recently termed *synthetic metabolism* [1], aims to unlock the potential chemical space available to microorganisms, enabling the production of entirely new compounds and even the design of pathway variants towards the same target product [2, 3, 4]. Identification of the best choice of target chemicals and biotechnological systems is not trivial, however. In reality, some bio-catalysts are more efficient than others, even when they have been designed to produce the same target chemical [5, 6]. An ability to accurately predict the designs likely to be superior would minimize experimental testing and hence optimize the use of available resources.

For any complex biological system, it is difficult to determine the most important factors, and parameters thereof, that influence catalytic performance. It is widely understood, however, that glucose catabolism inevitably results in the phosphorylation of ADP and AMP to form ATP, and the reduction of electron carriers (e.g. $NAD^+$ and $NADP^+$). These co-factors are subsequently hydrolysed and oxidized, respectively, when carbon source(s) is converted into biomass and by-products. For the sake of simplicity, we refer to these metabolic events hereinafter as 'production' and 'consumption' of ATP (also called 'energy') and NAD(P)H (also called 'redox'), respectively. Co-factor recycling is essential to allow central carbon metabolism to continue, i.e. to enable homeostasis [7]. The metabolic system that results in co-factor balance and robustness to environmental changes has evolved to facilitate survival of the species, and not to act as a host for biocatalysis serving human objectives. Hence, it is not surprising that an organism with an introduced synthetic metabolic pathway does not have optimal co-factor balance. Such an imbalance in the production and consumption of redox and/or energy by the engineered target pathway will result in the dissipation of co-factors by native metabolic processes such as cell maintenance and waste release, or promotion of growth over bioproduction. Metabolic waste products are therefore indicators of imbalances in the metabolic network, compromising the overall efficiency of the biocatalytic conversion of carbon towards the target [8].

In fact, even small changes in co-factor pools can have wide effects on metabolic networks and bio-production [9], and the inability of engineered systems to reach homeostasis can lead to partial or even full disruption of the cell's physiological state [5, 8, 9]. In order to maximize the ability to engineer optimal bio-catalysts, it is therefore essential to identify the constraints that pathway-specific co-factor imbalances may impose on the wider metabolic network.

Several articles have discussed this topic. For example, de Kok *et al.* [10] suggested a *positive yield theory* whereby optimal biosynthetic pathway flux is more likely when there is small, positive ATP excess remaining after the target pathway has utilised what it needs, enabling some but limited biomass production to keep the culture alive. Notably in this case, the authors referred to 'pathway' as the entire metabolic network of the cell. To support this argument, they compared the low ethanol yield from a high-biomass producing *Saccharomyces cerevisiae* strain, against the higher ethanol yield in a low-biomass producing *Zymomonas mobilis* strain. They argued that the difference was due to excess ATP production by the *S.cerevisiae* strain and that ATP surplus caused too much cell growth, burdening metabolism and decreasing product formation [10].

Albeit with a different terminology and reference point, Dugar and Stephanopoulos reached a similar conclusion based on a theoretical framework that assessed the imbalance of metabolic pathways and the effect this has on theoretically optimal product yield [5]. Using stoichiometric and energetic calculations, they quantified the relative potential of synthetic pathways, concluding that the most effective equilibrium between substrate and product optimization was found in fully balanced (net zero) or ATP-requiring (negative ATP yield) pathways. In this article, 'pathway' referred to the leading route towards target production from a central carbon metabolite, not the entire metabolic network of the cell. Their calculations facilitated a comparison between different pathway yields after adjusting for any imbalances, providing insights into where the imbalance in question may be occurring and an adjusted theoretical yield estimate. This information can then be used to select better performing pathways and guide engineering strategies to render the pathway more balanced and thus more yield-efficient [5]. However, their approach is built on a set of case-specific and not easily generalizable assumptions, does not consider various experimental conditions or biological settings, nor does the method scale up to larger metabolic networks or address the implications of pathway imbalance at the genome scale. Given that an understanding of co-factor metabolism is very useful and informative to predict the superiority of biosynthetic pathways, but the method published by Dugar and Stephanopoulos suffers from a lack of flexibility, we asked whether it would be possible to integrate both pathway-specific and network-specific balance assessments and carry out a similar analysis but using a more transferable and easy-to-implement computational framework?

The principal aim of the present study was therefore to implement a co-factor balance analysis (CBA) protocol to quantify the co-factor balance of metabolic engineering designs using well known constraint-based modelling techniques, such as Flux Balance Analysis (FBA) [11, 12], parsimonious FBA [13], and MOMA [14]. Using a stoichiometric model of *Escherichia coli* [15] and a series of different butanol production pathways as case studies [16, 17], CBA was used to evaluate how variations in ATP and redox demands contribute to yield efficiency. The study highlighted the impact of the underdeterminacy of FBA, demonstrated by considerable dissipation of excess ATP and NAD(P)H in high-flux futile cycles. Although some futile cycling may take place naturally, we assumed that their activation would not turn on and off as easily due to internal regulation, insufficient enzyme quantities and/or thermodynamic constraints imposed by both the chemistry of each reaction and *in vivo* metabolite concentrations [18]. Two methods to reduce high flux futile cycles were attempted, resulting in the formation

of biomass with 7 out of 8 engineered models, even when biosynthetic production was set as the objective function for optimization.

The CBA protocol helped explain why some pathways resulted in higher yields than others. Furthermore, both FBA and the approach developed by Dugar et al. [5] reached similar theoretical yield values and agreed on the highest yielding pathway. However, they differed in the way co-factor imbalances are adjusted both at the ATP and NAD(P)H level.

## Results and discussion

### Modified core stoichiometric models of *E.coli*

Eight synthetic pathways for the production of butanol and butanol precursors were selected for this study due to their distinct energy and redox requirements (Fig 1A). To enable target production *in silico*, we introduced reactions corresponding to the same engineered pathways as originally proposed and experimentally implemented by Menon *et al.* [16] and Pasztor *et al.* [17] into the *E.coli* Core stoichiometric model [15], resulting in a total of eight models, which we will refer to as BuOH-0, BuOH-1, tpcBuOH, BuOH-2, fasBuOH, CROT, BUTYR, BUTAL (Table 1).

Fig 1B summarizes the theoretical carbon yields and ATP and NAD(P)H coefficients of each synthetic pathway. For convenience, these stoichiometric coefficients are referred to as ***co-factor demand*** (negative values, indicating that the co-factor is consumed by the introduced pathway) and ***co-factor surplus*** (positive values, indicating co-factor production by the introduced pathway). Whilst all butanol pathways have the same redox demand but vary in ATP demand, the butanol precursor pathways have no ATP demand but instead vary in redox demand (Table 1).

Maximal product yield estimates were obtained by selecting the corresponding sink reaction as the objective function and maximizing these using parsimonious FBA (pFBA) [13]. For the wild type model, growth rate optimization was selected as the objective function. Under aerobic conditions, carbon yields ranged between 59.94–66.67% (refer to S2 and S3 Tables for pFBA-calculated fluxes), within the range of reported carbon yields for butanol calculated using alternative methods [5]. Under anaerobic conditions, the range increased to span 35.14–66.67%. It became noticeable, however, that the butanol models with highest ATP demands (tpcBuOH through to fasBuOH) had lower target production efficiencies. We suspected that these differences may stem from the need to utilise oxidative PPP to supply additional redox and the recycling of AMP and ADP, since these were not accounted for by the calculations presented in Dugar et al. [5].

The solution space of all models was investigated by Flux Variability Analysis (FVA) [19] (S1 Table). Considering only co-factor related reactions, a single solution was found with the wild type and models tpcBuOH, BuOH-2 and fasBuOH. In contrast, the butanol producing models BuOH-0 and BuOH-1 had varying flux ranges in 14 out of 85 and 18 out of 87 reactions, respectively. Notably, none of the reactions displaying multiple solutions were directly in the path towards butanol, and all were involved in futile co-factor cycles (see Fig 4).

### Co-factor Balance Assessment (CBA) produces distinct energy and redox profiles under aerobic and anaerobic conditions

The Co-factor Balance Assessment protocol (CBA) was designed to track co-factor production across the metabolic network and predict contributions to biomass, waste, target production and metabolic maintenance in each model. The goal was to use CBA to inform how co-factor properties of target pathways and the host cell influence yield efficiencies and facilitate an

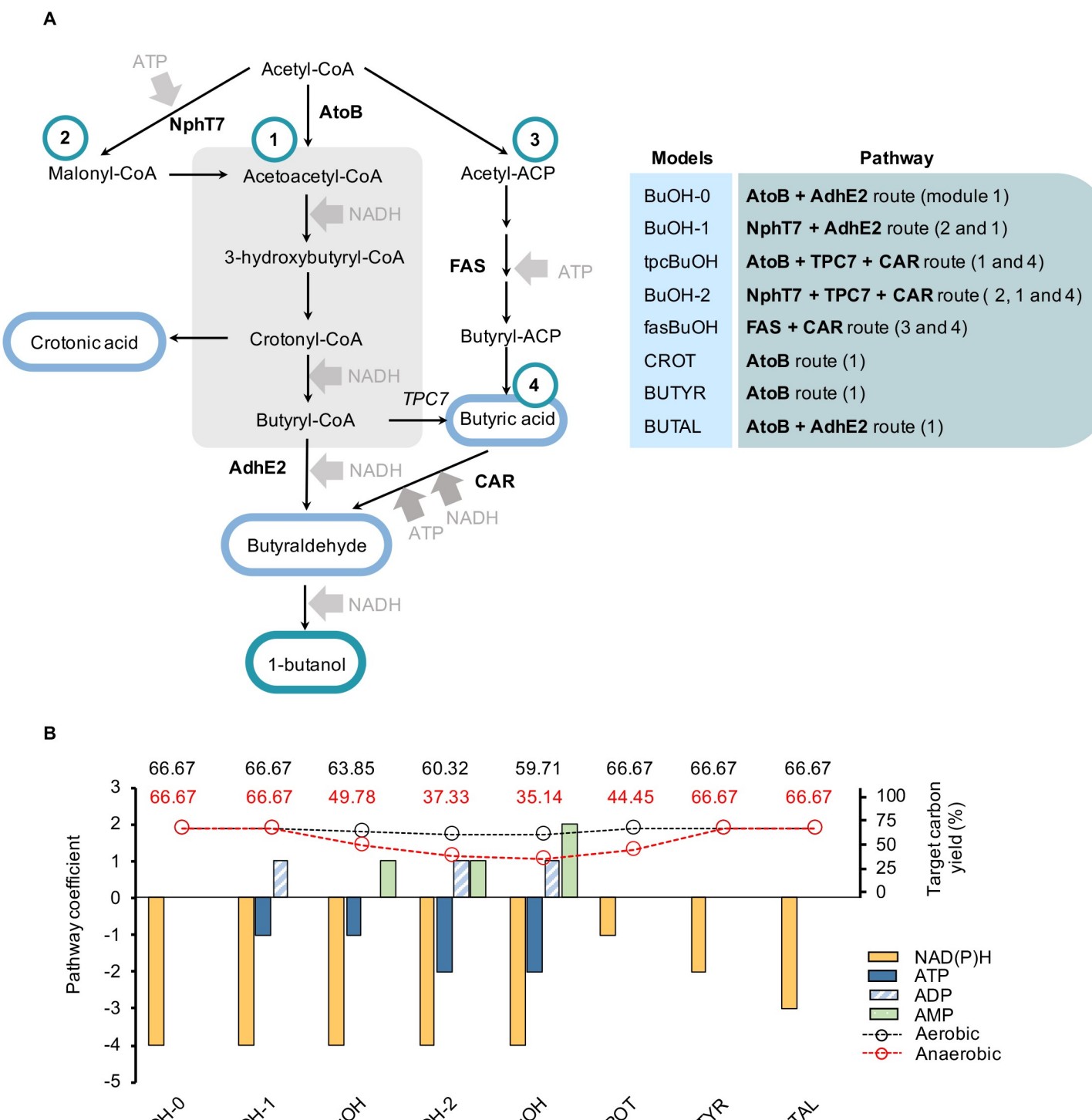

**Fig 1. Engineered pathways used in this study and their co-factor requirements.** Eight pathways that produce butanol (dark blue circle) and butanol precursors (light blue circles) were selected and introduced into the *Escherichia coli* Core Model to yield the stoichiometric models used in this study. These pathways are based on variations of the so called 'Core Pathway' (module A, grey), which is redox dependent and ATP neutral. By combining these modules, 8 unique pathways with varying demands for ATP and redox are possible. **(B)** Co-factor requirements of all pathways introduced into the *E.coli* Core model to simulate butanol and butanol precursor production, and the aerobic (black) and anaerobic (red) carbon yields are shown as a percentage of glucose carbon influx after target production maximization. Co-factor requirements are calculated as the sum of stoichiometric coefficients in all reactions starting from acetyl-CoA through to the final target molecule. Negative ATP/NAD(P) H coefficients represent co-factor demand, which refers to the consumption of a particular co-factor by the introduced pathway, indicating ATP/NAD(P)H going into the

reaction. Co-factor surplus, alternatively, is used to describe any co-factor being produced or released by a pathway. NAD(P)H surplus is indicated as positive NAD(P)H released by the pathway (subsequently from NAD(P) going into the reaction). CP–Core Pathway; ACP–acyl carrier protein; AtoB–acetyl-CoA acetyltransferase; AdhE2 – aldehyde alcohol dehydrogenase; NphT7 –acetoacetyl-CoA synthase; TPC–acyl-ACP thioesterase; CAR–carboxylic acid reductase.

understanding why some pathways perform better than others, from a co-factor usage perspective. CBA is a COBRApy-compatible Python protocol that sums the energy and redox synthesis fluxes of all reactions involving each of the two co-factors, and divides it into four categories: (1) biomass production, (2) product production, (3) waste release and (4) cellular maintenance (Fig 2). For example, a strain of *E.coli* engineered to produce butanol diverts a particular amount of energy and redox to produce the chemical target, whilst the rest is distributed across reactions that lead to biomass formation, metabolic maintenance and waste release. Upon linear optimization, the CBA protocol determines the net flux through each category, and this information can be used to understand how effective an engineered system is at producing a chemical target, with respect to the resources being dissipated to achieve the optimal objective.

The CBA calculation was applied to all eight models with introduced butanol or butanol precursor product pathways (hereafter 'engineered' models) under both aerobic (Fig 3A) and anaerobic (Fig 3B) conditions with target product excretion set as the objective function. These models were compared to the equivalent ATP and redox profiles of the wild type model (WT) when optimized for maximal biomass yield. Under aerobic conditions, solutions for the engineered models displayed smaller magnitude fluxes for ATP synthesis and consumption than WT (Fig 3A), in line with a lower requirement for ATP by the product pathways given that the backbone route towards butanol production is ATP neutral. In the absence of $O_2$, however, the ATP production levels were similar for all models apart from BuOH-2 and fasBuOH, which also presented the lowest yields (Fig 3A). Under both aerobic and anaerobic conditions, solutions for the engineered models showed no biomass accumulation. For BuOH-0, one of the highest yielding butanol models, more than half of the generated ATP went into the waste category, specifically being burned via the ATPM reaction (S4 and S5 Tables). All butanol models relied on the glyceraldehyde-3-phosphate dehydrogenase and pyruvate dehydrogenase reaction (PDH) for the supply of redox. The PDH reaction is known to provide the extra redox needed for butanol production [20]. As illustrated in Fig 2, PDH is labelled as 'waste', because NADH formation contributes to the loss of carbon through $CO_2$ release (hence the positive value observed for NAD(P)H waste under aerobic conditions,

**Table 1. Summary of key features of the modified *E.coli* models used in this study.** We have indicated model names, alongside their introduced reactions, target chemical, corresponding objective function (as per reaction ID), total number of model reactions and metabolites, and also the ATP and NAD(P)H pathway coefficients, calculated as the sum of reaction stoichiometry coefficients of all introduced reactions from acetyl-CoA to the final target. CP—Core Pathway.

| Model name | Introduced pathway | Target | Objective Function | Metabolites | Reactions | Degrees of freedom | ATP | NAD(P)H |
|---|---|---|---|---|---|---|---|---|
| WT | | biomass | biomass | 63 | 77 | 14 | | |
| BuOH-0 | AtoB + CP + AdhE2 | butanol | BTOH_sink | 70 | 85 | 15 | 0 | -4 |
| BuOH-1 | NphT7 + CP + AdhE2 | butanol | BTOH_sink | 72 | 87 | 15 | -1 | -4 |
| tpcBuOH | AtoB + CP + TPC7 | butanol | BTOH_sink | 71 | 86 | 15 | -1 | -4 |
| BuOH-2 | NphT7 + CP + TPC7 | butanol | BTOH_sink | 73 | 88 | 15 | -2 | -4 |
| fasBuOH | Butyryl-ACP route (FAS) | butanol | BTOH_sink | 77 | 91 | 14 | -2 | -4 |
| CROT | AtoB + CP | crotonic acid | CROAC_sink | 68 | 83 | 15 | 0 | -1 |
| BUTYR | AtoB + CP | butyrate | BTAC_sink | 69 | 84 | 15 | 0 | -2 |
| BUTAL | AtoB + CP | butyraldehyde | BTAL_sink | 69 | 84 | 15 | 0 | -3 |

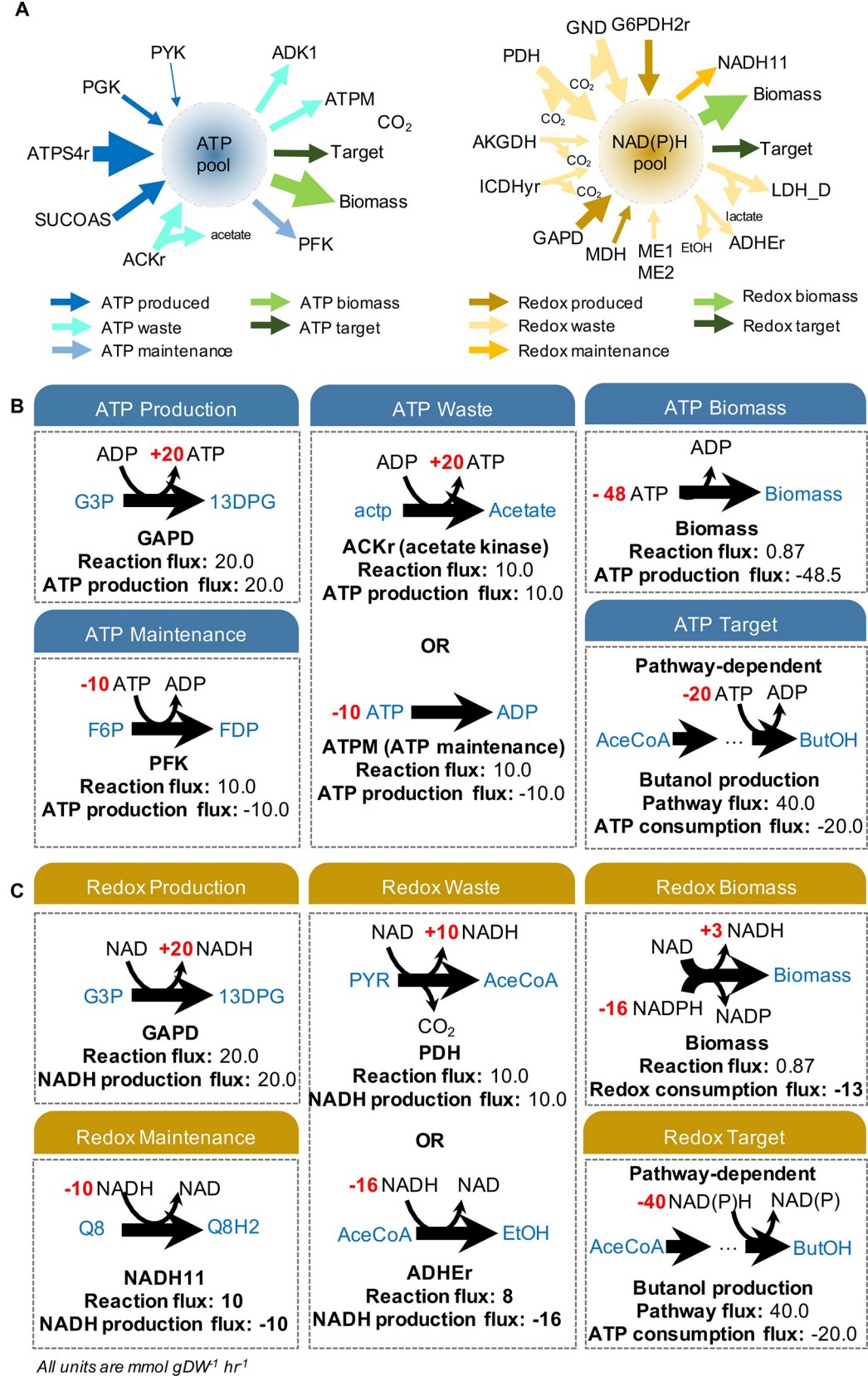

*All units are mmol gDW$^{-1}$ hr$^{-1}$*

**Fig 2. Toy illustration of ATP and NAD(P)H reactions and reaction categories accounted for by the CBA protocol.**
**(A)** All reactions in the *E.coli* Core Model that directly contribute to the intracellular levels of ATP and NAD(P)H pools (blue or yellow circle, accordingly). Arrows pointing inwards on the left display reactions leading to ATP or NAD(P)H build-up (i.e. co-factor production), while arrows pointing outwards on the right show reactions that drain the co-factor pools (i.e. co-factor consumption). The thickness of the arrows represent the varying fluxes of these reactions. The CBA protocol identifies all co-factor related reactions producing and/or consuming ATP or NAD(P)H, it records their fluxes and distributes them across five core categories: (1) co-factor production, (2) biomass production, (3) waste release, (4) cellular maintenance and (4) target production (this category is target product specific). **(B)** Theoretical example of how the classification of ATP reactions is handled by the CBA protocol. Co-factor fluxes (here illustrated by the varying arrow thickness) are dependent on the co-factor stoichiometric coefficient and flux calculated by FBA. **ATP production** accounts for all reactions that generate a positive ATP flux. The **ATP waste** category accounts for both ATP produced during acetate production, but also ATP consumed in ATP-hydrolysing reactions (also known as 'ATP burning' reactions), such as ATPM and ADK1. **ATP biomass** includes the ATP flux consumed during biomass formation. The **ATP target** category is pathway-specific, accounts for only those synthetic reactions introduced into the stoichiometric model, and will lead to a positive or negative flux according to whether the synthetic pathway leads to the formation or drain of intracellular ATP, respectively. If the synthetic pathway is ATP-neutral, the net value for this category will be zero. **ATP maintenance** includes any ATP consumed in additional metabolic activities and not considered in the aforementioned categories. **(C)** Theoretical example of how the classification of redox reactions is handled by the CBA protocol, similarly to (B). The NAD(P)H waste category also accounts for reactions GND, PDH, AKGDH, ICDHyr, which produce NAD(P)H but simultaneously release $CO_2$, and reactions such as LDH_D and ADHEr that consume NAD(P)H and release fermentation products. For categories including both positive and negative co-factor fluxes, the net is calculated for that category. Figure design inspired by [38].

Fig 3A, yellow). However, it simultaneously supplies the target pathway a key limiting factor, NADH, which is essential to optimize flux towards butanol production.

All metabolic pathways with the same end-product will have the same net redox requirements unless there is a change in non-target products (e.g. fermentation products, biomass). However, we observed considerable variation in NAD(P)H categories between models, both under aerobic and anaerobic conditions. Examining individual reactions (S2 and S3 Tables and S1 Fig), we noticed that the TPC route in models tpcBuOH, BuOH-2 and fasBuOH included a carboxylic acid reductase reaction that consumes 1 mol NADPH and produces 1 mol AMP from ATP, causing the coupling of electron metabolism with energy metabolism. As a result, we observed (1) 18.8%, 18.2% and 23.3% of total NAD(P)H was produced by the PPP resulting in a higher yield of NADPH per glucose and (2) activation of the ADK1 reaction to recycle AMP. Even though the butanol pathways all have the same demand for electrons, they have differing requirements for ATP. Homeostatic adjustments to the different ATP requirements resulted in changes in metabolism influencing also NAD(P)H. Moreover, although the flux through PPP was lower under anaerobic conditions relative to aerobic for models BuOH-2 and fasBuOH, flux through PPP surprisingly increased for tpcBuOH under anaerobic conditions. Further differences in the CBA redox profiles may arise from the fact that flux may or may not be directed via co-factor-dependent routes (e.g. PFL (electrons channelled into $H_2$ or excreted formate under anaerobic conditions) *vs*. PDH (electrons channelled back into $NAD^+$ as per S2 and S3 Tables), a concept known as 'degeneracy' or 'genetic buffering', brought by identical reactions coded by different genes that constitute alternative yet functionally overlapping pathways [21].

More generally, it was also observed that in order to cater to the increasing demands for ATP across the butanol pathways, the systems simply produced more net ATP, as depicted by the steady increase in ATP production along the x-axis (e.g. compare BuOH-2 with BuOH-0 on Fig 3B, blue). In contrast to these observations, the butanol precursor models (CROT, BUTYR and BUTAL), which did not demand ATP and only partly involved the Core Pathway, simply produced less ATP and also less NAD(P)H. We asked ourselves, is bacterial metabolism really this flexible? I.e. is the range of flux solutions predicted by stoichiometry-based modelling greater than what is possible in reality?

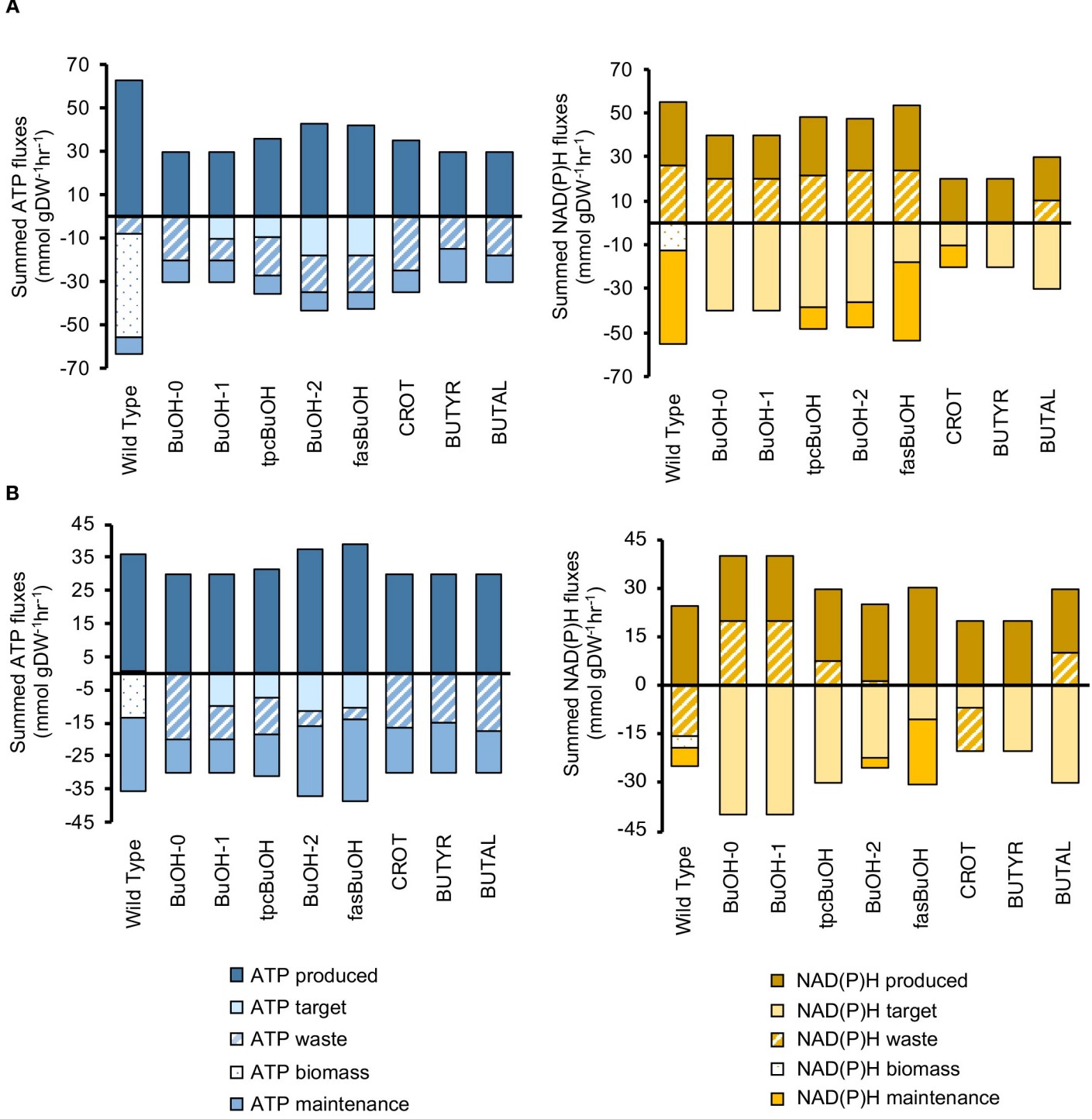

**Fig 3. CBA-derived network co-factor usage profiles.** After FBA optimization, the COBRA-based CBA protocol classifies ATP and NAD(P)H-related reactions according to whether these co-factors were consumed or produced during biomass, waste, target production or cellular maintenance. All models were initially unconstrained and simulated under both aerobic and anaerobic conditions. **(B)** ATP and NAD(P)H profiles under aerobic conditions; **(C)** ATP and NAD(P)H profiles under anaerobic conditions.

## The CBA pipeline highlights the underdeterminacy of FBA through co-factor dissipation

The general metabolic cost for non-growth-associated energy requirements, ATPM (Eq 1), is represented by an artificial reaction that breaks down ATP into ADP and Pi:

$$ATP \rightarrow ADP + Pi \tag{1}$$

In contrast to the wild type (measured at 7.6 mmol gDW$^{-1}$hr$^{-1}$ as per [22]), all models displayed a flux increase through either the ATPM or ADK1 reaction of up to 3-fold, ranging between 7.6–25 mmol gDW$^{-1}$ hr$^{-1}$ under aerobic conditions (S2 Table). Up to 71.4% of the total ATP produced was dissipated through these reactions alone, suggesting surplus energy in many of the models. The ATP neutrality of the Core Pathway (Fig 1A) could be causing the net ATP excess in these systems, as ATP is generated by substrate-level phosphorylation during glycolysis in order to produce acetyl-CoA, the primary precursor for target production. This results in an increased need to hydrolyse ATP in models including pathways with low or no ATP demand. As the ATP demand of the synthetic pathways increased, the fraction of ATP wasted by ATPM, or other ATP-burning futile cycles (Fig 4A), also gradually dropped (Fig 3A). The observation that artificially enforced ATP-hydrolysis can enhance product yield with engineered *E. coli* [23] supports the idea that ATP availability influences the allocation between biomass and other carbon-products.

Even under anaerobic conditions, where most models produced similar amounts of ATP, the fraction of ATP wastage ranged between 13% to 66.6% (Fig 3B), indicating that models with no or low ATP demands still dissipated surplus energy through ATP-burning reactions or cycles (described further in the following section). We also noticed that any redox imbalances in models tpcBuOH, BuOH-2 and fasBuOH were circumvented by the activity of NAD(P) transhydrogenase THD2 (Eq 2):

$$THD2: \quad NADH + NADP + H^+ \rightarrow H^+ + NAD + NADPH \tag{2}$$

The non-growth-associated dissipation of excess ATP, also referred to as 'energy spilling' or 'ATP burning', has been proposed as a principle for cells to handle energy surplus [24], but the extent to which *E.coli* does so is less understood [25]. The reversible nature of ATP synthase has also been suggested through the action of the rotational mechanism of the F1 subunit, but only under stress conditions [26]. In the case of redox balance, transhydrogenase activity is also known to be one of the various mechanisms to guarantee redox homeostasis [27]. Some fermentative bacteria can also alter their net ATP production as they change their end products [24]. However, given the limited understanding of *E.coli*'s capabilities to dissipate surplus energy [25], and earlier reports suggesting the flexible nature of stoichiometric models including synthetic pathways that are co-factor imbalanced [28], these co-factor burning observations were suggestive of FBA having more flexibility than what would be expected in reality. In contrast, based on observations from fermentation studies, the analysis by Dugar et al. assumed that the cell achieves energy and redox homeostasis through biomass and glycerol formation, respectively [5]. Left to its own devices, FBA did not resort to such solutions with the core wild-type model.

## Manual constraints of co-factor futile cycles leads to yield-efficient and biomass-viable solutions

During the assessment of our case studies, we asked whether constraining the observed flexibility of FBA would result in more realistic flux distributions. The first option was manual

**A** High-flux ATP and redox cycling reactions

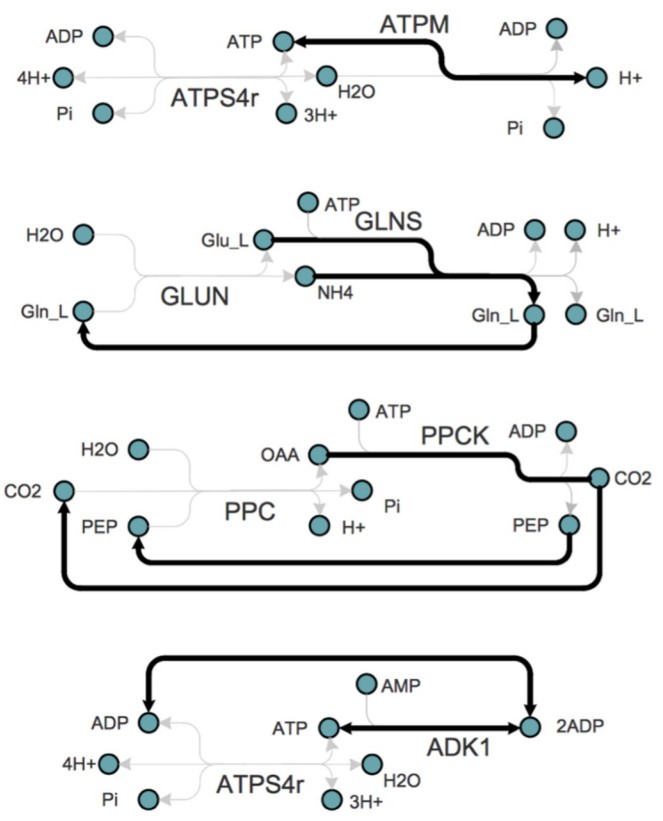 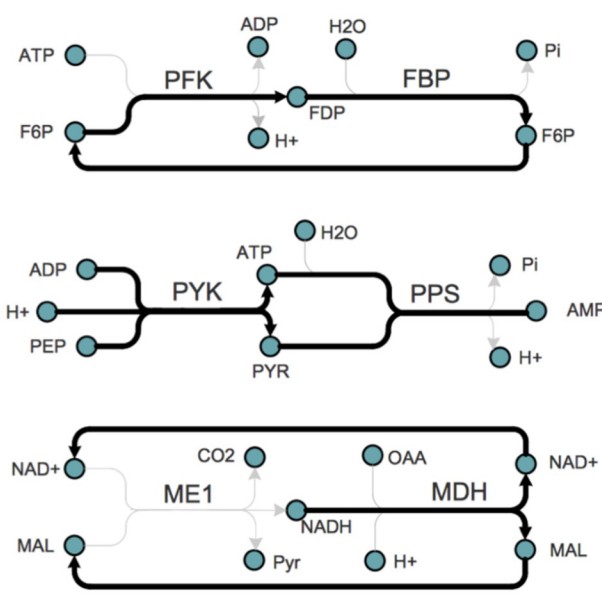

**B** Detection of high-flux futile cycle in FBA

**D** Optimization and detection of new futile cycle

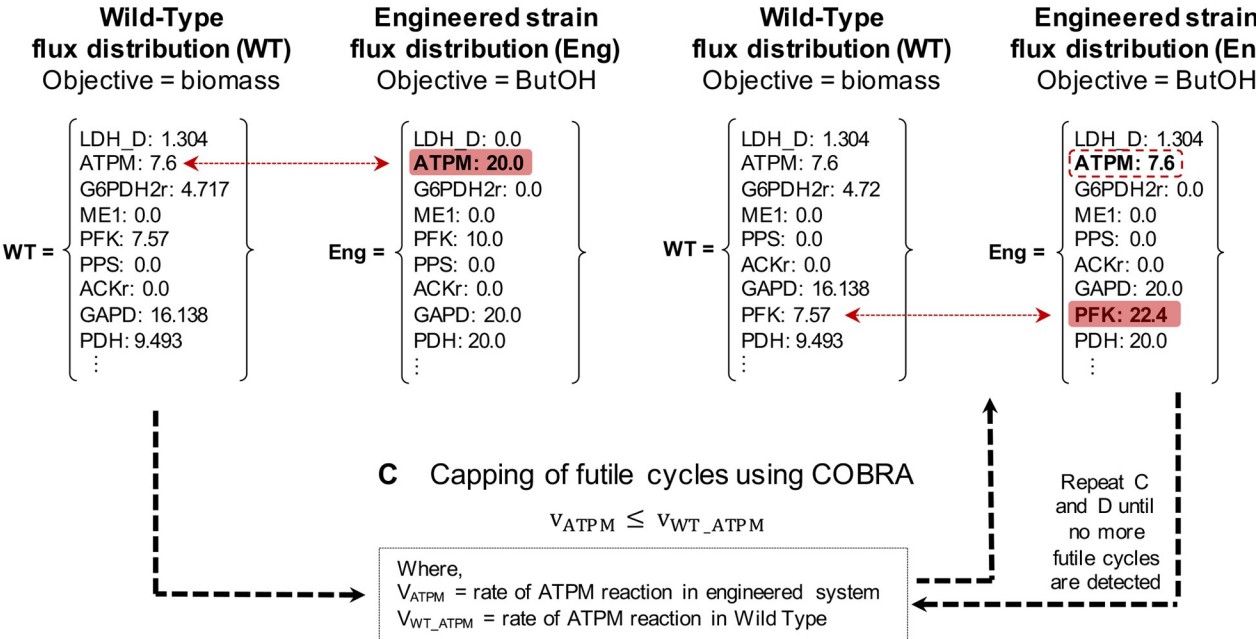

**C** Capping of futile cycles using COBRA

**Fig 4. Identification and removal of futile cycles. (A)** examples of ATP futile cycles identified in this study–pairs of cycling reactions in which ATP is consumed through one reaction and the original metabolites are recycled through the pair reaction. **(B)** ATP-burning and high-flux futile cycles were identified by directly comparing the engineered strain and the wild-type flux distributions. **(C)** The identified ATP-burning reaction or futile cycle was constrained by limiting the upper bound to the maximal flux observed for the equivalent reaction in the wild type. **(D)** After optimization, the flux distributions of the wild type and engineered system were compared and the next high-flux futile cycle would be detected and constrained accordingly (as per C). Steps (C) and (D) were repeated until no more futile cycles were detected.

correction, with the assumption that non-growth associated maintenance requirements reported by Varma & Palsson already captured the natural ATP dissipation levels that can occur in the *E.coli* metabolic network [22]. Consequently, we constrained the ATPM reaction of the engineered models to a maximum flux of 7.6 mmol gDW$^{-1}$ hr$^{-1}$, the value observed in the wild type when optimized for biomass formation.

When so doing, we noticed that the updated flux distributions (now ATPM-constrained) would instead divert the surplus energy through high-flux, co-factor spilling reaction pairs, also known as 'futile cycles'. Examples of identified high-flux, futile cycles are shown in Fig 4A. Futile cycles are pairs of anabolic and catabolic reactions that act in an antagonistic fashion, consuming either ATP or NAD(P)H through one reaction whilst phosphorylating or reducing a particular reactant, and a complementary reaction that regenerates the initial metabolite to close the loop [24]. Reactions like phosphoenolpyruvate carboxylase (PPC) and phosphoenol-pyruvate carboxykinase (PCK) can combine to form a futile cycle that potentially dissipates ATP [29, 30, 31], but this is highly likely to be conditional, as observed by Yang and colleagues when varying the dilution rate [29], or lead to an increase in biomass yield due to higher ATP production rather than less ATP turnover [32, 33]. Even when over-expressed, the potential antagonistic activity between pyruvate kinase (PYK) and phosphoenolpyruvate synthase in *E. coli* did not result in any significant futile cycle [34]. It has now become apparent that futile cycles are tightly regulated to prevent energy waste [24, 25].

The *in silico* cycles that dissipated the cofactor imbalance previously satisfied by upregulation of ATPM also involved the transhydrogenases THD2 and/or NADTRHD, and redox-driven reactions linking Glycolysis, PPP and the TCA cycles [35]. Like a whack-a-mole, with each constrained cycle appeared another. In a stepwise manner, new futile cycles were identified by directly comparing the flux distributions of the engineered models to that of the wild type (Fig 4B). After cycle detection, the non-cofactor-consuming reaction was capped by limiting its upper or lower bound according to the maximal flux value observed for the same reaction in the wild type (Fig 4C). This also meant that if the corresponding reaction was inactive in the wild type, the flux of the same reaction in the engineered system would be set to zero. This iterative, manual curation was repeated, followed by optimization and flux distribution evaluation until no more futile cycles were observed (Fig 4D). All reactions considered during manual constraining are included in S6 Table.

Biomass production competes for cellular resources against the biosynthetic pathways. In addition to energy or redox, biomass production also involves the synthesis of many other metabolites, so biomass production will reduce the maximum yield of butanol, which would in principle contradict the assumption of optimal yield under pFBA. However, the manually constrained models without any apparent futile cycles (Fig 5), and simulated to optimise target production, resulted in solutions that channelled excess co-factors through the biomass equation. Under aerobic conditions (Fig 5A), 7 out of 8 engineered models yielded solutions that led to both target and biomass accumulation. Under anaerobic conditions (Fig 5B), three of the models yielded solutions that were fully balanced both in terms of ATP and NAD(P)H through the use of fermentation product release only, and without any biomass production. Here, biomass production appears not to be contradicting the optimisation principle, but

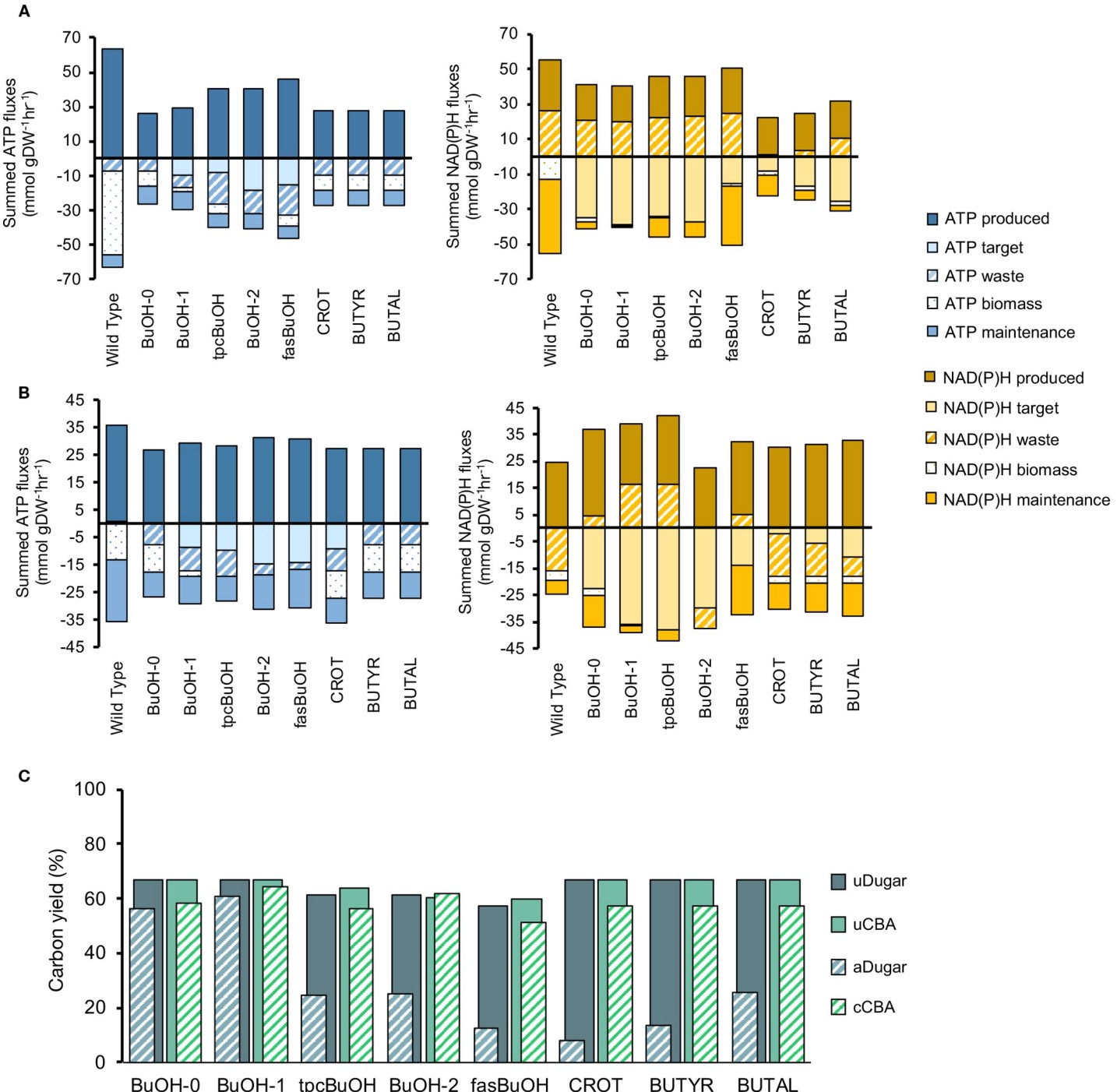

**Fig 5. CBA-derived co-factor usage profiles after manual curation of the models and comparison between CBA-derived estimates and estimates calculated using the method developed by Dugar et al.** The engineered models were manually constrained to minimize high-flux ATP futile cycles as described in the text, and led to the above **(A)** curated ATP and NAD(P)H CBA profiles under aerobic conditions and **(B)** ATP and NAD(P)H CBA profiles under anaerobic conditions. **(C)** Comparison of bioproduction carbon yields determined using the calculations developed by Dugar et al and FBA. uDugar–unadjusted Dugar-derived estimates; uFBA–unconstrained FBA (preliminary FBA results prior to applying any constraints); aDugar–Dugar-derived estimates after adjusting values according to ATP and NAD(P)H imbalances; cFBA–curated FBA yield estimates, obtained after manually constraining the flux distributions.

instead placing an upper limit on the maximum yield achievable, which is in line with Dugar & Stephanopoulos' suggestion that biomass is used as a sink for energy surplus to achieve metabolic balance, at the expense of product yield.

From a biotechnological perspective, an engineered organism having no ATP or redox flux consumed in a biomass reaction would be considered the most balanced, as all the key resources (ATP and redox) are utilised by the target biosynthetic pathway. In this respect, BuOH-1 appears to have the most balanced pathway after applying manual constraints as it has the highest carbon yield (64.43%) and lowest biomass yield. In reality, some biomass accumulation is clearly essential in all strains in order to synthesise the biocatalyst in the first place.

The tpcBuOH, BuOH-2 and fasBuOH models, which included butanol pathways with the highest ATP demands (Table 1), displayed an AMP imbalance in the TPC7 pathway, consequently lowering the yield to 56.33%, 61.33% and 51.07%, respectively. In these cases, 21%, 22.6% and 16.6% of the total ATP was used for AMP recycling through ADK1 (S1 and S7 Tables), so the metabolic network flux distribution required further readjustment to produce enough ATP for target production and maintenance, further altering the redox balance along the way. These results illustrate that the more intertwined the imbalance of a synthetic pathway is, the more the host needs to sacrifice a larger proportion of its energy budget to reach balance at the expense of product yield. This phenomenon was previously suggested by Weusthuis et al., who coined the concept 'incomplete oxidation' [36]. The methodology of Dugar and Stephanopoulos [5], which states that ATP-neutral or requiring pathways are more efficient, is only confined to pathway potential (i.e. it excludes a network-wide analysis) and does not account for the adjustment of additional co-factor imbalances. In contrast, the CBA protocol illustrates that a higher ATP (or redox) demand by the synthetic pathways may not always translate into higher productivity, because the host network then needs to accommodate the increased ATP demand by the synthetic pathway, with subsequent knock-on effects. For example, an imbalance in ATP homeostasis is typically solved by changes in the flux of pathways that involve electron transfer [8]. In contrast to previous studies that have studied the behaviour of co-factors in isolation [5, 37, 38], the CBA protocol strongly suggests that ATP and NAD(P)H balancing cannot be assessed in isolation from each other, or even from the balance of additional cofactors, such as AMP and ADP.

## FBA is an effective network-wide alternative to existing single-pathway cofactor balance assessments

We were interested to investigate whether CBA could provide a more complete balance assessment than existing methods. Using the calculations published by Dugar et al. [5], the biosynthetic capabilities of the butanol and butanol precursor pathways were calculated. These yield and efficiency metrics were obtained using the pathways' NAD(P)H demand, product release, and ATP, NADH and $CO_2$ surplus coefficients between their initial building block (i.e. acetyl-CoA), and their final target (S11 and S12 Tables). Final estimates are shown on Table 2.

Fig 5C displays a comparison between the above estimates (unadjusted pathway yield estimates, known as $Y^P$ in [5] but here labelled as uDugar, and adjusted estimates after accounting for redox, $CO_2$ and ATP imbalances, known as $Y^{P,G,X}$ in [5] but here labelled aDugar) and our CBA estimates (before and after constraining high-flux futile cycles), all normalized to the glucose uptake flux.

The method developed by Dugar et al. accounts for 2 net ATP and 4 NADH produced during glycolytic catabolism prior to target biosynthesis. Thus, this was accounted for in our calculations, given that the primary precursor for all pathways was acetyl-CoA. The available redox can then be directly used to produce 1 mol of butanol, or precursors thereof. Because

**Table 2. Maximum yield ($Y^E$ and $Y^{Ea}$), pathway yield ($Y^P$), adjusted pathway yields ($Y^{P,G}$ and $Y^{P,G,X}$) and pathway efficiency ($\eta$) of all butanol and butanol precursor pathways calculated using the stoichiometric and energetic calculations proposed by Dugar et al. [5].** Maximum yield ($Y^E$) is the maximum amount of product that can be produced from the substrate. $Y^{Ea}$ indicates the maximum yield in mol product/mol substrate. Pathway yield, $Y^P$, is pathway-specific and calculated from pathway stoichiometry. $Y^{P,G}$ is the adjusted pathway yield once any excess redox is depleted using an electron sink (i.e. glycerol). $Y^{P,G,X}$ is the adjusted pathway yield after any excess ATP is diverted towards biomass formation. $\eta$ is the ratio between $Y^{P,G,X}$ and $Y^E$.

| Pathway | $Y^E$ | $Y^{Ea}$ | $Y^P$ | $Y^{P,G}$ | $Y^{P,G,X}$ | $\eta$ |
|---|---|---|---|---|---|---|
| AtoB + AdhEr route | 1.000 | 0.411 | 1.000 | 1.000 | 0.844 | 0.844 |
| NphT7 + AdhEr route | 1.000 | 0.411 | 1.000 | 1.000 | 0.915 | 0.915 |
| AtoB + TPC7 route | 1.000 | 0.411 | 0.923 | 0.583 | 0.365 | 0.365 |
| NphT7 + TPC7 route | 1.000 | 0.411 | 0.923 | 0.571 | 0.379 | 0.379 |
| FAS + TPC7 route | 1.000 | 0.411 | 0.857 | 0.383 | 0.191 | 0.191 |
| AtoB route (CROT) | 1.333 | 0.637 | 1.000 | 0.395 | 0.161 | 0.121 |
| AtoB route (BUTYR) | 1.200 | 0.587 | 1.000 | 0.500 | 0.247 | 0.206 |
| AtoB route (BUTAL) | 1.091 | 0.437 | 1.000 | 0.667 | 0.415 | 0.381 |

the synthetic pathway introduced into model BuOH-0 has no ATP demand, there is a +2 ATP imbalance under this framework's assumptions. BuOH-1 is the most yield efficient solution both under CBA and Dugar frameworks. In line with their suggestions [5], this solution includes a synthetic pathway that is both ATP-requiring and diverts the least co-factor surplus towards biomass production (Fig 5A), yielding the highest butanol production while diverting the least amount of energy towards waste and/or futile cycles.

We noted a discrepancy in solutions from tpcBuOH, BuOH-2 and fasBuOH. Under the Dugar framework, these pathways exhibited an NADH surplus of up to 2 mol, so the maximal theoretical yield is penalized twice: it is first adjusted for any redox and $CO_2$ imbalance, then adjusted once more to account for any ATP imbalance. With CBA, however, both redox and ATP imbalances can be systematically addressed, as both co-factors are needed for the biomass reaction to carry any flux at all. Furthermore, the need for these solutions to recycle AMP and address NAD(P)H demand are handled by fine tuning the wider metabolic network to render both the pathway and the entire system at balance, so the impact on the final theoretical yield is lower. Here, the limitation of Dugar et al. becomes clear: they allow for only one possibility to address each potential imbalance: excess ATP can only be resolved via biomass production, whilst excess NADH is consumed by a glycerol sink. Similarly, Dugar et al. penalize models CROT, BUTYR and BUTAL because they present both ATP and redox surplus. In the case of fasBuOH, this analytical method cannot account for the pathway's dependence on fatty acid synthesis (FAS), so these reactions must be manually accounted for if we are to effectively consider the additional ATP and NAD(P)H requiring pathways ahead of malonyl-ACP production [39].

We conclude that, although both methodologies report similar unadjusted theoretical yields for all models, and both methodologies agree on the best performing pathway, the CBA protocol provides a more complete depiction on the metabolic potential of pathways and the limits they may pose on a biological network upon implementation. Recycling of co-factor by-products, co-factor maintenance reactions and the tight coordination between different subsections of metabolism are examples of very interesting observations we can make with the CBA protocol that we would otherwise be unable to account for under alternative methods.

## Evaluation of alternative methods to reduce futile co-factor cycles

In the present work, single linear optimizations were utilised, however the models were mostly underdetermined. We assumed that the more constraints from known biochemical principles that were added to the model, the narrower the range of phenotypes would be. Previously, it was shown that high-flux futile cycles and biomass limitations could be addressed through

manual curation of carefully selected reactions (Fig 4). In this section, we asked whether a similar outcome could be obtained by other approaches, including (1) 'loopless' FBA [40] (2) 'Minimization Of Metabolic Adjustments' (MOMA) method for predicting gene knock-out phenotypes [14] and by (3) constraining the model with flux data from $^{13}$C-MFA [41].

We first implemented the COBRApy 'loopless_solution' function, however, this did not eliminate the futile co-factor cycles (S2 Data File). Alternatively, we asked whether we could use flux data from Long et al. as lower and upper bound constraints in our models. Long et al. evaluated flux responses to 20 single-gene perturbations of upper central metabolic carbon metabolism in *Escherichia coli* [41]. Given that the stoichiometric model used for fitting their data is similar to the *E.coli* core model, and that the study involved direct knockouts of glycolytic reactions, this dataset provided a sound basis for comparison. First, *in silico* models of each single knockout evaluated in Long et al. were optimized using MOMA [14] with the wild type *E.coli* solution as a reference, capturing the flux of each CCM reaction measured by Long et al. This resulted in the generation of an *in silico* flux variability range (hereinafter referred to as MOMA range, S15 Table) for 15 reactions from CCM. This was compared to the ranges in flux variability observed for the same reactions in both wild type and knockout strains as determined by $^{13}$C Metabolic Flux Analysis (MFA) [41] (S13 Table) and FBA-based FVA [19] (S14 Table). Ten out of fifteen and twelve out of fifteen reactions displayed a greater variability range *in silico* (FVA and MOMA, respectively) compared to that measured with 13C-MFA (S13–S15 Tables). The MFA data indicated that some reactions were more "plastic" (i.e. more flexible, able to change flux more widely according to changes in demand), such as PFK, GAPD and PGK in glycolysis, whilst other reactions were more "rigid" (i.e. showing no or very little change in flux, such as ME1, ME2 and PPCK (ranges of 1.3)). Interestingly, some of the reactions that were rigid in reality were predicted to display wide flux ranges using both FVA (S1 Table) and MOMA (Fig 6A and S15 Table). These rigid reactions were also commonly involved in high-flux futile cycling in the unconstrained engineered models, very likely stemming from the underdeterminacy of the unconstrained models.

The MFA flux ranges were also implemented as upper and lower bound constraints for the same reactions in the engineered butanol models followed by optimization using the CBA algorithm (Fig 6B and 6C), in an attempt to evaluate whether this could minimize the need for manual capping of futile co-factor cycles. The use of MFA flux constraints was not sufficient to eliminate all co-factor dissipation in the engineered models, and MFA-based constraints did not result in any biomass formation. ATP burning through ATPM was still present in all solutions, with the exception of BuOH-2 and fasBuOH (S16 Table). If experimental constraints were also combined with manual capping of the ATPM maintenance reaction and FBP (and in the case of tpcBuOH also PPCK) following the manual curation procedure outlined earlier (Fig 4) (S17 Table), the predicted maximum target product yield was markedly reduced and in most models except for BuOH-0 and BuOH-1 they were similar to that predicted with the Dugar method (Fig 6B). We concluded that MFA-based constraints is not an ideal replacement for manual whack-a-mole futile cycle capping.

## Co-factor demand sensitivity analysis and the identification of "optimal" co-factor balance

The CBA analysis seemed to suggest that models having the least amount or no flux towards biomass are the most balanced ATP and redox-wise. For example, the manually curated BuOH-1 model had the least amount of futile cycling and ATP burning, the highest butanol yield and lowest biomass yield (Fig 5). Is this the optimum or could the catalytic system be improved even further? To better understand the effect of co-factor balancing on product

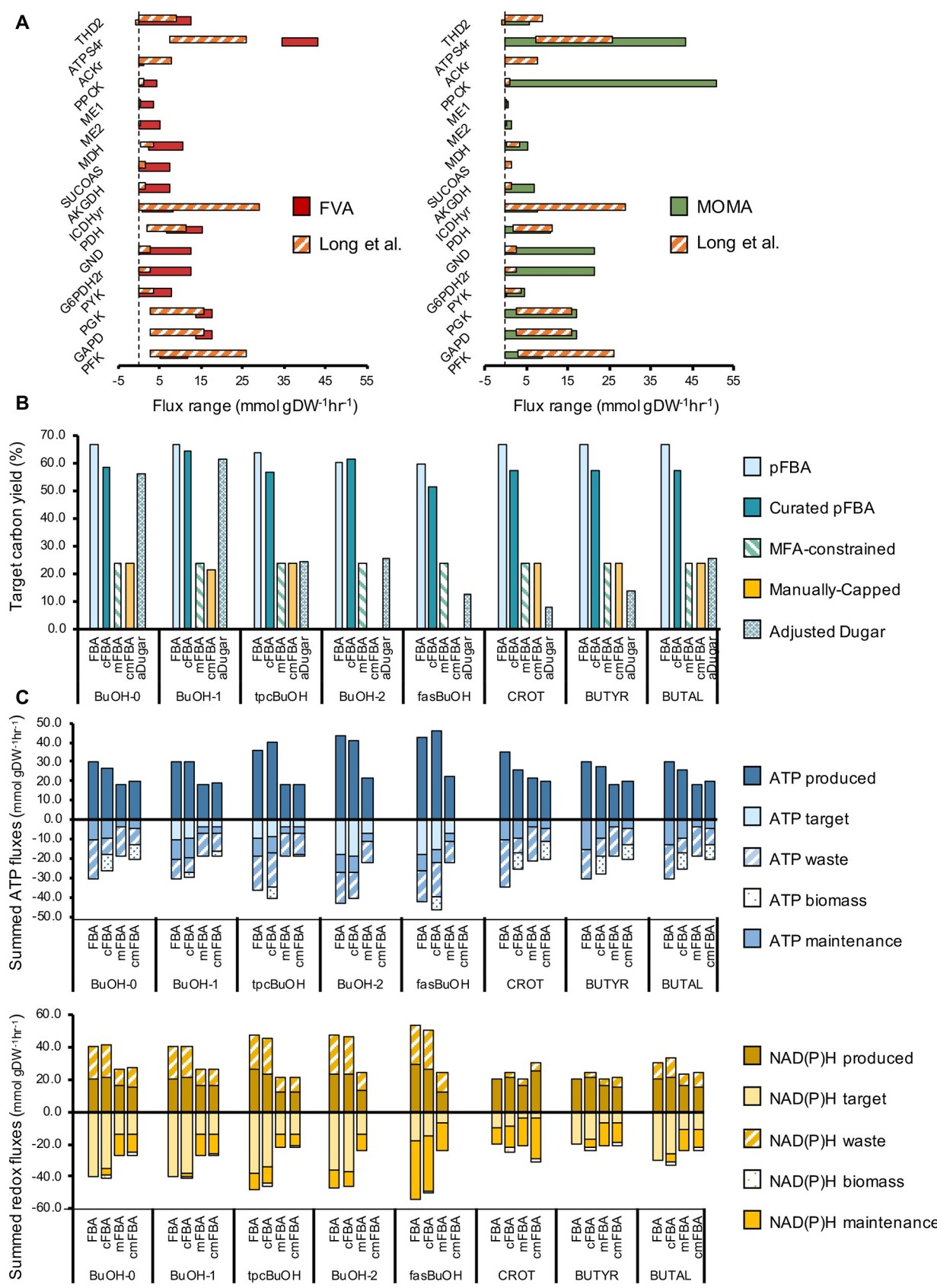

**Fig 6. MOMA compared to MFA-derived estimates, carbon yield efficiencies and CBA co-factor profile comparison across unconstrained, manually curated and experimentally constrained solutions. (A)** Flux ranges calculated with MOMA (green) and Metabolic Flux Analysis (orange stripes). MOMA ranges were estimated using the wild type solution as a reference and sequentially implementing the single-gene knockouts studied by Long et al. (2019) [46], with biomass formation as the objective function. MFA ranges were extracted from a pre-existing dataset (Long et al., 2019), using a Python algorithm to select the minimal and maximal flux ranges. **(B)** Carbon yields of butanol and butanol precursor models, compared across all approaches evaluated in this study: unconstrained pFBA (labelled 'FBA'); manually curated pFBA solutions with minimized high-flux futile cycling (labelled 'cFBA'); experimentally-constrained solutions using MFA-derived flux data (labelled 'mFBA'); experimentally-constrained solutions using MFA-derived flux data with further capping in co-factor cycling reactions (labelled 'cmCBA') **(C)** ATP (blue) and NAD(P)H (yellow) CBA-derived cofactor usage profiles compared across all approaches evaluated in this study (labels identical as previously).

yield, a sensitivity analysis was carried out in which the demand for ATP and NAD(P)H was artificially varied. Such an approach could also potentially be used to generate a metric indicative of the co-factor imbalance in each model. The co-factor sensitivity analysis was applied to the manually-curated, aerobic, butanol models BuOH-0, BuOH-1, tpcBuOH and fasBuOH, by introducing an artificial NADH and ATP sink/generator that modifies the pathway's ATP and NADH stoichiometric coefficients, to simulate pathways with both co-factor surplus (excess ATP/NADH produced by the target pathway) and co-factor demand (ATP/NADH going into the pathway).

The resulting 3D landscapes describe the impact of changes in co-factor demand on product yield under aerobic conditions (Fig 7). Under aerobic conditions, models BuOH-0 and BuOH-1 could withstand growing pathway ATP demand and redox surplus by forming a plateau at a theoretical carbon yield of 66.67% (Fig 7A and 7B), a behaviour associated with more robust systems [42]. These two pathways exhibit high glycolytic flux under aerobic conditions and the increasing demand for ATP is satisfied by higher respiration thanks to the increasing redox availability. The original synthetic pathways in these two engineered systems reached carbon yields of 58.53% and 64.47%% (Fig 7), but optimal co-factor ratios boosted the yield by 8.14% and 2.2%, accordingly. The highest biomass rates were recorded in the presence of both growing ATP and NADH surplus, at the expense of butanol production.

The tpcBuOH and fasBuOH models displayed limited capability to accommodate any change in co-factor demand, thus forming a cliff and causing a drop in product yield (Fig 7C and 7D). Both of these CAR-dependent models have high ATP demand, as AMP needs to be recycled via an ATP-consuming ADK1 reaction. They also have a high requirement for NADPH, resulting in increased flux through the Pentose Phosphate Pathway. As so referred in [42], these more unstable models (tpcBuOH and fasBuOH) achieved carbon yields of 56.43% and 51.1% respectively, without the artificial co-factor variation, but by manipulating their co-factor demands the maximal carbon yields increased up to 61.3% and 56.67%, accordingly. These observations suggest that if the sweet spot for optimal balance between the introduced pathway and host metabolism is small, it reduces the chances that a high-yielding integrated combination of pathway and host cell metabolic network can materialise. It doesn't exclude the possibility of high yield, but it makes it less likely.

These results suggest that it would be possible to determine a stoichiometrically "optimal" ATP/NAD(P)H ratio for each pathway. Knowing this, pathway engineering can be theoretically guided both in terms of selecting the optimal host strain background and by indicating the co-factor robustness of the pathway and likelihood that high yield is achieved. This information opens up a new horizon for further metabolic engineering adjustments that can potentially lead to more rapid implementation of high-yielding production strains.

## Conclusions

This study presents a stoichiometric-modelling-based Co-factor Balance Assessment (CBA) protocol to monitor co-factor usage and its system-wide impact on cell behaviour and the

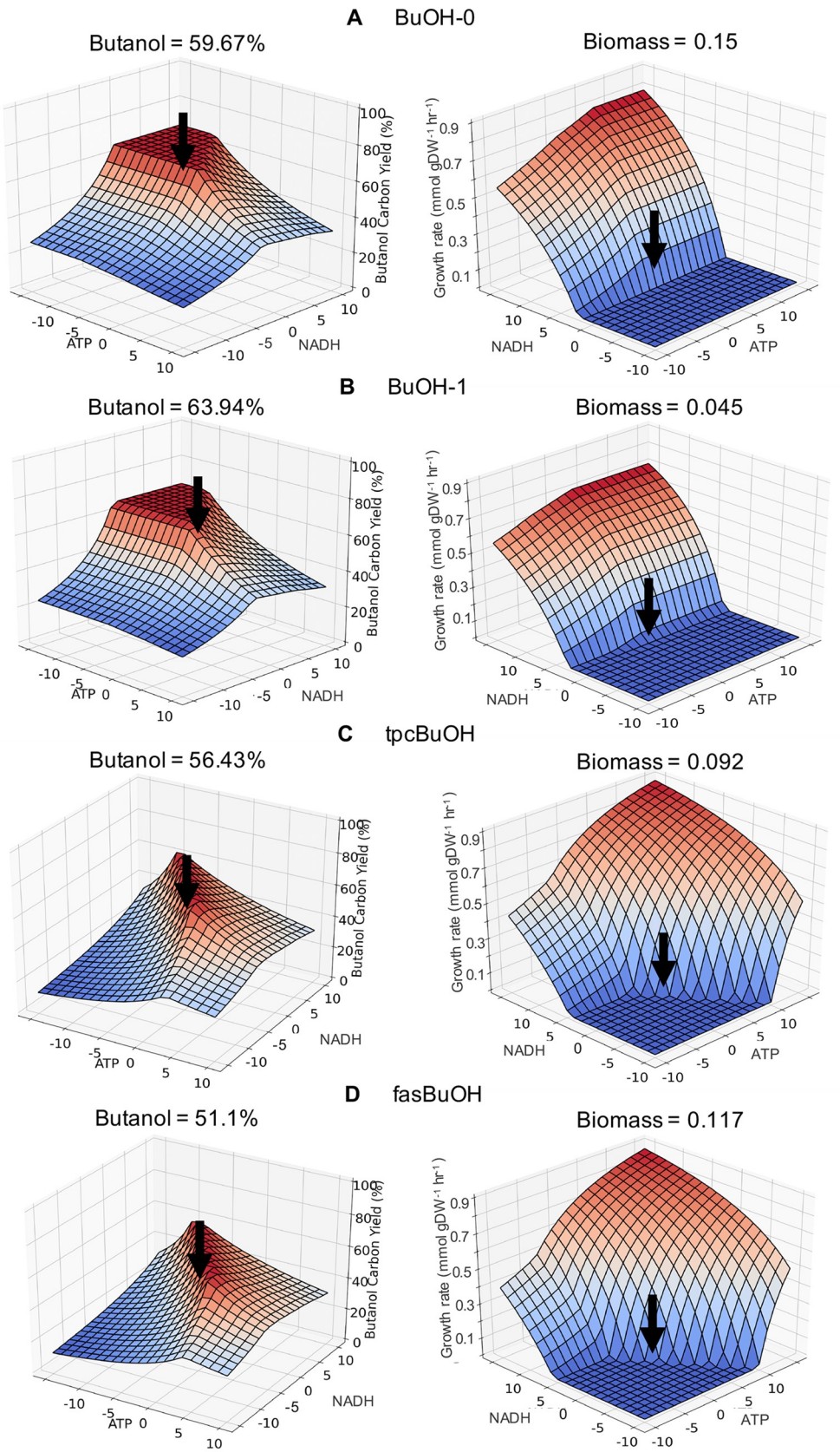

**Fig 7. butanol carbon yield (%) and biomass production rates (mmol gDW$^{-1}$hr$^{-1}$) of engineered *E.coli* strains in response to changes in ATP and NADH demands.** Each model represents a unique pathway variant for butanol production, which has been manually curated and optimized for the selected objective under aerobic conditions. (A) BuOH-0, comprised of route AtoB + AdhE2; (B) BuOH-1, including reactions NphT7 + AdhE2; (C) tpcBuOH, made up of AtoB + TPC7; (D) fasBuOH, comprising reactions NphT7 + TPC7.

design of optimal catalysts. With this, metabolic engineering designs can be selected based not only on the highest yield achievable but on knowledge of the stress imposed by co-factor metabolism. Our CBA protocol describes co-factor (im)balance as the fraction of co-factor diverted to biomass, maintenance or waste, instead of target production, and captures how organisms may be limited by pre-existing redox and energy constraints.

Our results suggest that introducing cofactor-balanced pathways reduces the burden placed on the rest of the metabolic network. They also indicate that ATP and NAD(P)H balance cannot be assessed in isolation from each other, or from the balance of additional cofactors, such as AMP and ADP. We evaluated two means whereby FBA models could be constrained to reduce their flexibility. Even though manually constrained solutions had no apparent futile cycles, it is vital that we remain prudent and do not forget FBA's flexible nature and optimistic estimates when using this *in silico* approach.

We also provide insights into the yield and co-factor profiles of optimally balanced strains through a co-factor sensitivity analysis. Identifying the optimal balance is especially relevant when narrowing down pathway variants, as identification of the most efficient and robust pathway for target production becomes an essential step in the design approach. Our results indicate that we can substantially increase target production if we modify the ATP and redox demands of our introduced pathways. This information opens up a new horizon for further metabolic engineering adjustments that can lead to better yields, whereby a synthetic build-up of NAD(P)H/ATP, or alternatively a synthetic sink for NAD(P)H/ATP could be introduced into the system to cater to the network-wide co-factor profile.

While our analysis uses butanol and butanol precursors as proof-of-concept, our CBA pipeline thrives on its ability to evaluate different stoichiometric models, target products, pathway routes, strains, carbon sources and environmental and genetic conditions. It becomes a powerful way to inform engineers how to achieve optimal strain designs, as having knowledge of the extent of co-factor imbalance can more accurately discriminate engineered systems that are more balanced (and thus more productive) and indicate potential genetic manipulations that will lead to the design of more efficient strains.

## Materials and methods

All work described in this study was done using the Constraint-Based Reconstruction and Analysis toolbox for Python (COBRApy version 0.13.3) [43] and Gurobi solver (version 5.5.0) [44]. All scripts and functions extensively used in our analysis were run in the Python 3 environment (version 3.7.4) [45].

### Target catalysts

All simulations employed an *Escherichia coli* Core Model [15]. In order to enable butanol or butanol precursor production in *Escherichia* coli, the following synthetic pathways were implemented into separate copies of the *E.coli* Core Model, to yield stoichiometric models iDAG85, iDAG87, iDAG86, iDAG88, iDAG91, iDAG83, iDAG84_butyr and iDAG84_butal. We assembled 8 synthetic models and analysed a total of 9 models: (1) *iDAG85 (*referred to as BuOH-0 in this study)*, a butanol producer, includes the combination of reactions AtoB and AdhE2

along with the so-called Core Pathway (CP) shown in Fig 1. It comprises a total of 85 reactions and 70 metabolites and is ATP neutral; (2) *iDAG87* (referred to as BuOH-1) produces butanol via the ATP consuming reaction NphT7 [46]. BuOH-1 includes 87 reactions and 72 metabolites; (3) *iDAG86* (referred to as *tpcBuOH)*, a butanol-producing pathway that integrates enzymes AtoB and converts butyryl-CoA into butyraldehyde via a thioesterase and an ATP-dependent carboxylic acid reductase reaction (referred to as TPC7 in Menon et al. [16]). This model is made up of 86 reactions and 71 metabolites; (4) *iDAG88*, hereinafter known as *BuOH-2*, which incorporates reactions NphT7 and TPC7 and includes 88 reactions and 73 metabolites; (5) *iDAG91 (*referred to as *fasBuOH)*, an ACP-dependent butanol pathway that relies on Fatty Acid Synthesis (FAS), a thioesterase to release butyric acid and an ATP-dependent carboxylic acid reductase to generate the aldehyde. It is comprised of 91 reactions and 77 metabolites; (6) *iDAG83* (labelled *CROT* in this study*)*, which produces crotonic acid and is made up of 83 reactions and 68 metabolites; (7) *iDAG84_butyr* (known as *BUTYR)*, a butyrate producer including 84 reactions and 69 metabolites; (8) *iDAG84_butal* (labelled as *BUTAL)*, which yields butyraldehyde and is made up of 84 reactions and 69 metabolites; and finally (9) *Wild Type*, or *WT*, the version of the *E.coli* Core Model that excludes all reactions required for butanol production and fatty acid biosynthesis, containing a total of 77 reactions and 63 metabolites.

All reactions were added as per COBRApy standards and have been detailed in S19 Table. All engineered models included target-specific production, transport and sink reactions (reactions that drain the final product out of the metabolic network) [8]. Models have been provided in SBML format and may also be replicated by running the file S1 Code in the specified python environment.

## Flux Balance Analysis (FBA)

All models were simulated using Parsimonious FBA (pFBA) for computing optimal phenotypes [13]. pFBA is a bi-level optimization method that minimizes the total sum of flux whilst optimizing for the selected objective using FBA. Net flux is minimised subject to optimal biomass as follows:

$$min \sum_{j=1}^{m} v_{irrev,j}$$

$$s.t. \max v_{objective} = v_{objective,lb}$$

$$s.t. S_{irrev} \times v_{irrev} = 0$$

$$0 \leq v_{irrev,j} \leq v_{max}$$

Where $m$ = number of irreversible reactions in the network; $S_{irrev}$ = stroichiometric matrix; $v_{irrev}$ = non-negative, steady-state flux; $v_{objective}$ = approximates the theoretical objective; and $v_{objective,lb}$ = lower bound of the objective rate. This is followed by the maximization of target per unit flux, by optimizing the ratio of the objective to the square of the total network flux:

$$\max \frac{v_{objective}}{\sum_{i=1}^{n} v_i^2}$$

$$s.t. Sxv = 0$$

$$v_{min} < v_i < v_{max}$$

Modified models were optimized for the drain of butanol or butanol precursor, whilst the wild type had maximal growth rate selected as the optimization principle.

All models were initially unconstrained. Prior to optimization, the model was pre-processed to set the primary carbon source, glucose, constrained to a maximum intake rate of -10 mmol gDW$^{-1}$ hr$^{-1}$. Constraints necessary for computational minimal media conditions were set as default [47]. The presence or absence of oxygen was also modulated to run our simulations under both aerobic and anaerobic conditions. In aerobic simulations, the oxygen uptake rate was set to a maximum of 10 mmol gDW$^{-1}$hr$^{-1}$. Alternatively, oxygen uptake was constrained to zero under anaerobic conditions.

## Co-factor Balance Assessment (CBA) pipeline construction

The algorithm developed for *co-factor balance assessment* (CBA) was written as a Python function, using the COBRApy package. It was built as a single function, found in file S2 Code, and can be called out to calculate the sum of all energy and redox synthesis fluxes and divides it into four categories:

- Biomass production: energy and redox consumed during biomass formation

- Target production: total energy and/or redox consumed or produced during target optimization (this is target-specific)

- Waste release: total energy burned to produce ADP/AMP, or energy and redox consumed during the release of $CO_2$ or fermentation of by-products.

- Cellular maintenance: energy and redox consuming reactions which are not accounted for in the previous categories

Our CBA protocol relies on the following inputs: a stoichiometric model, a matching flux distribution and a list of target reactions needed for target optimization. After FBA (or pFBA) optimization, CBA uses the stoichiometric model as a source of co-factor stoichiometry information. It matches the identified reactions to their corresponding flux estimate to determine whether ATP or NAD(P)H is being either produced or consumed, by calculating a co-factor flux score (CFS):

$$CFS_{i,j} = S_{i,j} \, x \, v_j$$

where; $S_{i,j}$ i = stoichiometry of co-factor i (ATP or NAD(P)H) in reaction j; $v_j$ = reaction flux

For net ATP and NAD(P)H production (Net ATP or NAD(P)H produced within CCM), all positive CBSs are summed up. Then, the co-factor fluxes corresponding to each category are calculated and adjusted as per Fig 8. The final scores consist of summed flux values that describe the overall "weight" of each category.

The CBA protocol relies on the following assumptions. CBA solely tracks ATP and NAD(P)H metabolism, while all other co-factors are excluded from the analysis. For modelling purposes, it assumes that NADH and NADPH are interchangeable, even though in reality NADH and NADPH are not biologically equivalent. We assume in this work that the above categories are the main categories classifying co-factor metabolism, whilst other co-factor assisted biological functions such as intracellular and extracellular transport, cell motility, cell division, stress, gene and protein ex- pression, are excluded from analysis (or believed to be considered within the maintenance category, if at all).

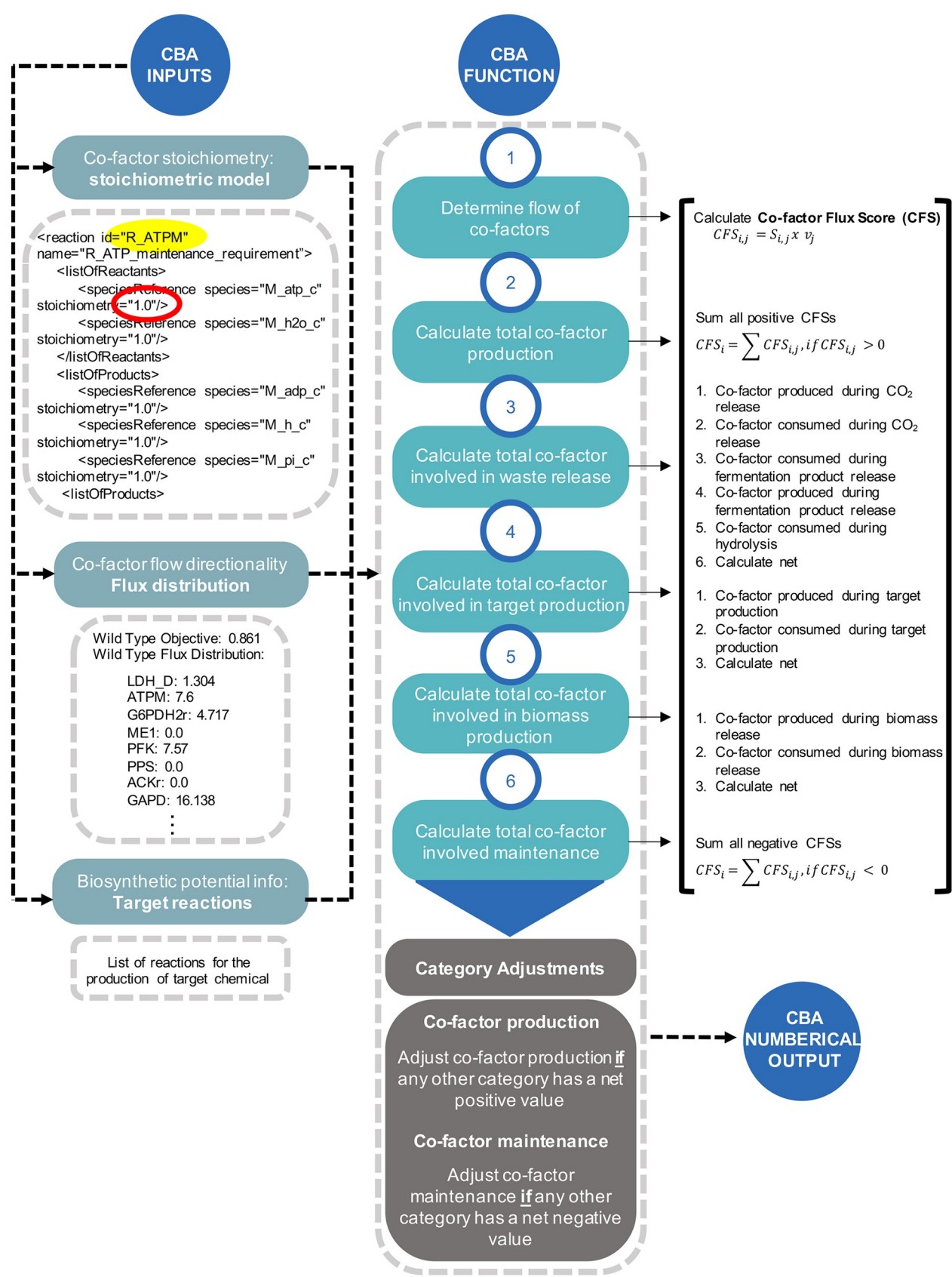

**Fig 8. Components of Cofactor Balance Assessment (CBA) pipeline and summarised workflow.** A stoichiometric model, flux distribution and a list of target reactions are required to call the CBA function in the python environment. Stoichiometric models contain reaction information, such as whether they consume or produce ATP and NAD(P). We used the *E.coli* Core stoichiometric model and the COBRApy package, and selected reactions were implemented to build the path to novel products. CBA classifies reactions in the model according to whether they are involved in the consumption or production of NAD(P)/ATP, assigns them a cofactor balance score, and groups them into categories as represented above. Finally, the total balance per category is calculated the total sum of flux and adjusted to provide a final value for each category. The result is a profile displaying the fraction of the total cofactor produced involved in maintenance, biomass, target and waste production.

CBA simulations reported in this study may be replicated by running S3 Code.

## Flux Variability Analysis (FVA)

Flux variability analysis was used to calculate the minimum and maximum allowable flux values for each reaction involving ATP and/or NAD(P)H within the Core stoichiometric model of *E.coli* [19]. The feasible range of reaction fluxes by maximization and minimization of each reaction flux was calculated at 99% of the maximal value of the objective function. The mathematical formulations for minimization and maximization problems are show below:

Maximization:

$$\max v_j$$

$$\text{Subject to } S_{ij} \times v_j = 0$$

$$c^T v = Z_{objective}$$

$$v_j^{min} \leq v_j \leq v_j^{max}$$

Minimization:

$$\min v_j$$

$$\text{Subject to } S_{ij} \times v_j = 0$$

$$c^T v = Z_{objective}$$

$$v_j^{min} \leq v_j \leq v_j^{max}$$

Where $Z_{objective}$ is the value of the objective function. If $n$ is the number of fluxes, then 2n LP problems are solved under FVA.

## Implementation of constraints derived from 13-C labelling experiments

Fluxomic data derived from $C^{13}$-labelling experiments by Long et al. was used [41].

## Experimental data pre-processing

Prior to use, their original dataset was formatted as follows: First, (i) only those reactions present in the Core stoichiometric model of *E.coli* were kept, and families of reactions (e.g. pfka and pfkb, which are represented by a single PFK reaction in the stoichiometric model) were averaged to provide for a single reaction estimate; and (ii) the reference data (WT data) was averaged to include a single flux value per reaction and. A total of 15 reactions remained in the set. Finally, (iii) the data was normalized according to the recorded glucose uptake rates.

## Constraint implementations derived from experimental datasets

The formatted Long dataset (S1 Data File) was used as an input to generate upper and lower bound constraints. This dataset includes fluxes measured for selected central carbon metabolism reactions (shown as rows), in a series of knockout strains, where the knocked-out reactions are shown as columns. We assumed that for each reaction measured in this study, their corresponding fluxes in each strain represented the catalytic range for that particular reaction. Using a python script algorithm to extract the maximal and minimal flux values recorded for each reaction, we built a database (a python dictionary) of minimal and maximal values, which were then implemented as upper and lower bound constraints for the respective reaction in the metabolic network. Implementation of constraints may be replicated by running S4 Code in the specified python environment.

## Development of MOMA ranges

To compare the MFA data, we simulated each knockout strain in Long et al. using Minimization of Metabolic Adjustment (MOMA) [14], using the wild type solution as reference. MOMA searches for a vector x that presents minimal distance from a given flux vector L so that the Euclidean distance is minimized as:

$$\min f(x) = L(x) + \frac{1}{2} x^T Q x$$

Where L = vector of length N; Q = NxN matrix defining the linear and quadratic part of the objective function; $x^T$ = transpose of x; x = vector with minimal Euclidean distance from L

For each simulated knockout strain, the reactions corresponding to the same reactions estimated in Long et al. were recorded. This produced a reaction flux range equivalent to that derived using the experimental data.

Results may be replicated by running S4 Code in the specified python environment.

## Cofactor ratio analysis

To improve the intracellular redox and energy status for butanol production (by increasing and decreasing the availability of NAD(P)H and ATP), we created a landscape analysis that assessed the system's behaviour under pathway-specific (i) NAD(P)H and ATP overproduction, (ii) NAD(P)H and ATP consumption, and (iii) no redox or energy generation.

This was achieved y including an artificial ATP production step (Eq 4) into the existing stoichiometry of the aldehyde reductase reaction (Eq 3), the final step in the butanol production chain. Next, we looped through the reaction's A(X)P and NAD(P) stoichiometries in a loop that was followed by pFBA optimization, to produce a landscape recreating metabolic scenarios from cofactor surplus (Eq 5), to no cofactor usage (Eq 6), through to cofactor demand (Eq 7). This method can be replicated by running the "cofactor_ratio_analysis.py" file in the specified python environment.

$$butyraldehyde + NADH + H^+ \rightarrow n - butanol + NAD \tag{3}$$

$$butyraldehyde + NADH + H^+ + ADP + Pi \rightarrow n - butanol + ATP + NAD \tag{4}$$

$$butyraldehyde + \boldsymbol{X}\ NADH + \boldsymbol{X}\ H^+ + \boldsymbol{Y}\ ADP + \boldsymbol{Y}\ Pi \rightarrow n - butanol + \boldsymbol{X}\ NAD + \boldsymbol{Y}\ ATP \tag{5}$$

$$butyraldehyde + \boldsymbol{0}\ NADH + \boldsymbol{0}\ H^+ + \boldsymbol{0}\ ADP + \boldsymbol{0}\ Pi \rightarrow n - butanol + \boldsymbol{0}\ ATP + \boldsymbol{0}\ NAD \tag{6}$$

$$butyraldehyde + X\ NAD + Y\ ATP \rightarrow n-butanol + Y\ ADP + Y\ Pi + X\ NADH + X\ H^+ \quad (7)$$

For our purposes, we evaluated a range of -10 to 10 (consumption of -10 ATP/NAD(P) to a co-factor surplus of +10). Every time the stoichiometry changed, we optimized for the selected objective (butanol production).

Results may be replicated by running S5 Code in the specified python environment.

## Implementation of Dugar & Stephanopoulos' method

The calculations published by Dugar and Stephanopoulos [5] were applied to the synthetic pathways under study (**Fig 1**), assuming a carbon feedstock of glucose and a glycolytic fermentative process. Briefly, first, an energetic calculation of **maximum yield** ($Y^E$) was used to assess the maximum amount of product that can be produced from the substrate, measured in moles of product per mol of substrate (Eq 8). Next, a system of linear equations (Eqs 9–13), where stoichiometric coefficients *a*, *b*, *c*, *d* and *e* (NADPH release, product yield, ATP release, NADH release and $CO_2$ release) were estimated for each catalyst and normalized per glucose carbon atom, were used to then calculate pathway yield (*YP*, Eqs 14 and 15). *YP* was later adjusted to account for pathway inefficiencies. Firstly, assuming that cells thrive to be redox-neutral, any excess NAD(P)H is balanced using an electron sink (glycerol production) via Eq 16, to yield *YP,G*. Finally, excess ATP is diverted towards biomass formation, using Eq 17. After these adjustments, the final yield value, *YP,G,X* represents the highest possible yield, assuming there are no competing pathways (Eq 18).

Ultimately, Pathway efficiency (η) is calculated by doing the ratio of *YP,G,X* and *YE* (Eq 19). Our target catalysts comprise more than one chemical reaction, so the Dugar method was applied by calculating the net balance of ATP, redox agent and *CO2* release for all carbon-carrying reactions from the original carbon source (glucose) to the end product.

$$Y^E = \frac{Y_S}{Y_P} \quad (8)$$

$$v_1 = -CH_2O - aNADPH + bProduct + cATP + dNADH + CO_2 \quad (9)$$

$$v_2 = -CH_2O + 2NADPH + CO_2 \quad (10)$$

$$v_3 = -CH_2O + 4.82ATP + CO_2 \quad (11)$$

$$v_4 = -CH_2O - \frac{1}{3}ATP - \frac{1}{3}NADH + CH_{\frac{8}{3}}O\ (glycerol) \quad (12)$$

$$v_5 = -CH_2O - \frac{\alpha}{1+\varepsilon}ATP + \frac{1}{1+\varepsilon}CH_{1.83}O_{0.56}N_{0.17}(Biomass) + \frac{x}{1+\varepsilon}NADH + \frac{\alpha}{1+\varepsilon}NADH + CO_2 \quad (13)$$

$$Y^P = Y\frac{v_1}{v_1 + v_2 + v_3} \quad (14)$$

$$Y^P = Y \frac{1}{1 + a/2 - (c/4.82)|_{if\ c<0; else c=0}} \tag{15}$$

$$Y^{P,G} = Y \frac{v_1}{v_1 + v_2 + v_3 + v_4} = Y \frac{1}{1 + a/2 + \left((d - c/4.82)|_{if\ d-c<0; else d-c=0}\right) + 3d} \tag{16}$$

$$x = \frac{4(1 + \varepsilon - \kappa)}{2} \quad \text{(where } \kappa = 4.2 = \text{degree of biomass reductance)} \tag{17}$$

$$Y^{P,G,X} = Y \frac{v_1}{v_1 + v_2 + v_3 + v_4 + v_5} = Y \frac{(a + x)}{(1 + a/2)(a + x) + c(3x + 1 + \varepsilon) + d)3a - (1 + \varepsilon)} \tag{18}$$

$$\eta = Y^{P,G,X} / Y^E \tag{19}$$

## Supporting information

**S1 Table. Flux Variability Analysis of the Wild Type and engineered models using the** *Escherichia coli* **Core Model.** Used 100% of optimum and optimized for biomass formation or production of butanol, crotonate, butyric acid or butyraldehyde, accordingly. Minimal and maximal range units are in mmol gDW$^{-1}$ hr$^{-1}$. Highlighted in grey–reactions presenting variability ranges, instead of unique fluxes.
(DOCX)

**S2 Table. pFBA flux distributions of wild type and engineered models under aerobic conditions.** Used the *Escherichia coli* Core Model and parsimonious FBA for optimization, using either biomass formation or target production as the objective function, accordingly. Solutions were simulated under aerobic conditions (EX_o2_e_ = -10 mmol gDW$^{-1}$ hr$^{-1}$) and were otherwise unconstrained.
(DOCX)

**S3 Table. pFBA flux distributions of wild type and engineered models under anaerobic conditions.** Used the *Escherichia coli* Core Model and parsimonious FBA for optimization, using either biomass formation or target production as the objective function, accordingly. Solutions were simulated under anaerobic conditions (EX_o2_e_ = 0 mmol gDW$^{-1}$ hr$^{-1}$) and were otherwise unconstrained.
(DOCX)

**S4 Table. CBA parameters and outputs of unconstrained models under aerobic conditions.** Reaction IDs, relevant co-factor and their stoichiometric coefficient, flux value, balance value and assigned balance category are included.
(DOCX)

**S5 Table. CBA parameters and outputs of unconstrained models under anaerobic conditions.** Reaction IDs, relevant co-factor and their stoichiometric coefficient, flux value, balance value and assigned balance category are included.
(DOCX)

**S6 Table. Candidate reactions for manual curation.**
(DOCX)

**S7 Table. CBA parameters and outputs of manually curated engineered models under aerobic conditions.** Reaction IDs, relevant co-factor and their stoichiometric coefficient, flux value, balance value and assigned balance category are included.
(DOCX)

**S8 Table. CBA parameters and outputs of manually curated engineered models under anaerobic conditions.** Reaction IDs, relevant co-factor and their stoichiometric coefficient, flux value, balance value and assigned balance category are included.
(DOCX)

**S9 Table. Flux Variability Analysis of the manually curated engineered models using the** *Escherichia coli* **Core Model under aerobic conditions.** Used 100% of optimum and optimized for butanol or butanol precursor production. Minimal and maximal range units are in mmol gDW$^{-1}$ hr$^{-1}$. Highlighted in grey–reactions presenting variability ranges, instead of unique fluxes.
(DOCX)

**S10 Table. Flux Variability Analysis of the manually curated engineered models using the** *Escherichia coli* **Core Model under anaerobic conditions.** Used 100% of optimum and optimized for butanol or butanol precursor production. Minimal and maximal range units are in mmol gDW$^{-1}$ hr$^{-1}$. Highlighted in grey–reactions presenting variability ranges, instead of unique fluxes.
(DOCX)

**S11 Table. Pathway coefficients for NADPH demand, product release, ATP release, NADH release and CO$_2$ release of all butanol and butanol precursor pathways.** Reaction-specific stoichiometric coefficients were calculated per reaction involved in the synthetic pathway, and the final pathway coefficient was retrieved as the net sum across all reactions. Respiro-fermentative conditions assumed that the release of 2 mol of acetyl-CoA yields 2 mol ATP, 2 CO2 and 4 mol NADH per mol of carbon source assimilated (glucose, C$_6$H$_{12}$O$_6$) prior to product formation. This was accounted for in the below calculations.
(DOCX)

**S12 Table. Normalized pathway coefficients a (NADPH), b (product), c (ATP), d (NADH) and e (CO$_2$) of all butanol and butanol precursor pathways.** Stoichiometric coefficients from S9 Table were normalized per carbon-mol of glucose.
(DOCX)

**S13 Table. Upper and lower bound constraints derived from 13-C labelled data.**
(DOCX)

**S14 Table. Upper and lower bound constraints derived from flux variability analysis (fraction of optimum = 0.972).**
(DOCX)

**S15 Table. Upper and lower bound constraints derived from MOMA.**
(DOCX)

**S16 Table. pFBA flux distributions of wild type and engineered models constrained using 13-C MFA data.** Solutions were simulated under aerobic conditions and optimized for their selected objectives using pFBA: the wild type was optimized for biomass formation whilst the engineered models were optimized for target production.
(DOCX)

**S17 Table. pFBA flux distributions of engineered models constrained using 13-C MFA data and additional manual constraining of high-flux futile reactions.** Solutions were simulated under aerobic conditions and optimized for target production using pFBA.
(DOCX)

**S18 Table. Reactions added to the Core Model of *E.coli* to implement biosynthetic production of butanol and butanol precursors.**
(DOCX)

**S1 Fig. pFBA flux distribution maps of unconstrained models under aerobic conditions.**
(DOCX)

**S2 Fig. Butanol carbon yield (%) and biomass production rates (mmol gDW$^{-1}$hr$^{-1}$) of engineered *E.coli* strains in response to changes in ATP and NADH demands under anaerobic conditions.** Each model represents a unique pathway variant for butanol production, which has been manually curated and optimized for the selected objective under aerobic conditions. (A) BuOH-0, comprised of route AtoB + AdhE2; (B) BuOH-1, including reactions NphT7 + AdhE2; (C) tpcBuOH, made up of AtoB + TPC7; (D) fasBuOH, comprising reactions NphT7 + TPC7.
(DOCX)

**S1 Model. Model iDAG85 (referred as BuOH-0 in this study, butanol producer, AtoB +AdhE2 route).**
(XML)

**S2 Model. Model iDAG87 (referred as BuOH-1 in this study, butanol producer, NphT7 +AdhE2 route).**
(XML)

**S3 Model. Model iDAG86 (referred as tpcBuOH in this study, butanol producer, AtoB +TPC7 route).**
(XML)

**S4 Model. Model iDAG88 (referred as BuOH-2 in this study, butanol producer, NphT7 + TPC7 route).**
(XML)

**S5 Model. Model iDAG91 (referred as fasBuOH in this study, butanol producer, FAS +CAR route).**
(XML)

**S6 Model. Model iDAG83 (referred as CROT in this study, crotonate producer).**
(XML)

**S7 Model. Model iDAG84_butyr (referred as BUTYR in this study, butyrate producer).**
(XML)

**S8 Model. Model iDAG84_butal (referred as BUTAL in this study, butyraldehyde producer).**
(XML)

**S1 Data. formatted MFA data from Long et al. used to derive upper and lower bound constraints.**
(XLSX)

**S2 Data. Loopless FBA results of butanol producers.**
(XLSX)

**S1 Code. Python script implementation of butanol and butanol producer models.**
(PY)

**S2 Code. Python script including functions developed in this study (CBA function included here).**
(PY)

**S3 Code. Python script for application of CBA function and whack-a-mole manual curation, under aerobic and anaerobic conditions.**
(PY)

**S4 Code. Python script for comparison between FVA, MFA and MOMA constraints, and application of constraints onto butanol product.**
(PY)

**S5 Code. Python script for analysis of co-factor rations.**
(PY)

## Author Contributions

**Conceptualization:** Laura de Arroyo Garcia, Patrik R. Jones.

**Investigation:** Laura de Arroyo Garcia.

**Methodology:** Laura de Arroyo Garcia.

**Supervision:** Patrik R. Jones.

**Writing – original draft:** Laura de Arroyo Garcia.

**Writing – review & editing:** Laura de Arroyo Garcia, Patrik R. Jones.

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
