## [Decision Letter · Decision Letter 0]

26 Feb 2020

Dear Dr Jones,

Thank you very much for submitting your manuscript "In silico  co-factor balance estimation using flux balance analysis informs metabolic pathway potential in Escherichia coli" for consideration at PLOS Computational Biology.

As with all papers reviewed by the journal, your manuscript was reviewed by members of the editorial board and by several independent reviewers. In light of the reviews (below this email), we would like to invite the resubmission of a significantly-revised version that takes into account the reviewers' comments.

While the data underlying the figures are available, the Python function for the co-factor balance assessment is missing. As this code and analysis is the main contributions of this study, we would propose that this be made available the reader.

We cannot make any decision about publication until we have seen the revised manuscript and your response to the reviewers' comments. Your revised manuscript is also likely to be sent to reviewers for further evaluation.

Sincerely,

Anders Wallqvist

Associate Editor

PLOS Computational Biology

Jason Papin

Editor-in-Chief

PLOS Computational Biology

While the data underlying the figures are available, the Python function for the co-factor balance assessment is missing. As this code and analysis is the main contributions of this study, we would propose that this be made available the reader.

Reviewer's Responses to Questions

**Comments to the Authors:**

Reviewer #1: In this paper, de Arroyo Garcia and Jones explore how cell metabolism can limit the yield of synthetic metabolic pathways. In the growing field of metabolic engineering, were cells are treated as potential “factories” to synthesize industrial compounds, it is important to consider the capability of the cells native metabolism to “mesh” with the demands of synthetic pathways, as these pathways will alter the homeostasis of cellular energy charge and redox.

The methodology is centered on the use of flux balance analysis, but with extensive manual curation of the reaction constraints. Throughout, various pathways to n-butanol, a non-native product in E. coli, are considered based on how their NADPH and ATP demands are met by the cell, and what the true pathway yields would be. The main technical novelty is a new script for tracking and categorizing how cellular ATP and NADPH are affected (produced or consumed) in the presence of the new pathway.

The concept is not altogether novel, as pointed out by the authors they are following concepts laid out by Dugar and Stephanopoulos 2011. However, this approach is useful, in that it uses a core metabolic model instead of a pathway in isolation. The authors are thorough as well, they systematically “hunt” and constrain futile cycles as well as key reactions that appear to be bounded when analyzing experimental MFA data of knockout strains. This latter addition is interesting, as the MFA data suggest that cellular metabolism is not as flexible in adapting to knockouts as it appears from stoichiometric metabolic models. As a result, the limitations of FBA are systematically revealed and addressed in order to arrive at a more accurate prediction of a pathway’s actual yield.

Overall the paper is well conceived and well-written. It is somewhat specific to butanol, but there is a variety of butanol pathways considered that allow somewhat general conclusions to be drawn about expected pathway yields based on ATP and NADPH usages. The technical novelty is a bit lacking.

Specific points that could be addressed to improve the manuscript:

The authors use FBA with product maximization, and then record rates through various ATP and NADPH reactions. FBA gives one particular flux distribution out of many, would it have been better to use FVA here instead?

Authors use product as cellular objective and then tabulate fluxes of ATP and NADPH-related reactions. However, there may be more realistic ways to estimate how the cell would respond to new pathways. The MOMA technique (Segre et al., 2002) is another way to estimate fluxes after network perturbation, such as addition of a biosynthetic pathway, while biomass remains the cellular objective. The authors could consider such alternative methods for comparison, to see if similar ATP and NADPH-consuming rates arise. Perhaps MOMA where a certain flux into production pathway is used as a constraint.

In Abstract and Introduction, authors refer to “flexibility” of computational predictions. Later in the manuscript it is clear they mean possible flux patterns to reach the same objective (FVA). I recommend defining what is meant by flexibility (perhaps plasticity?) as early as possible (preferably in Abstract).

The authors use various butanol pathways as case studies, and also attempt to generalize findings about the predicted yields of pathways with certain ATP and NADPH demands. The reliance on MFA data from multiple mutants could limit the approach to be only available for E. coli or yeast as host strains, but these are likely still useful to get an idea of how a synthetic pathway could be limited in different hosts.

From the review file and SI, it was not clear if all scripts and models have been made publicly available. Missing are the different models and the Python scripts used to perform CBA. I encourage the authors to provide those.

Looking at table 1, NADPH and NADH are lumped together when it comes to pathway requirements. How is this translated in the model? Is there one reaction for each cofactor? Please clarify.

Fig. 1 Could be helpful to have enzyme abbreviation directly on the figure

Fig. 2 A) What does the thickness of the arrows mean? Is it proportional to the flux amplitude? If yes, please state under which condition CBA was performed, with WT or one of the “engineered” models ?

B) Same as A), few flux values a represented but it is not clear where they come from

Fig. 2 and Fig.3 NADPH waste: In fig.2 caption, it says “waste category may also include reactions..” What are these other reactions?

l.248 “Under aerobic conditions, solutions for the engineered models showed no biomass accumulation” Was it also the case in aerobic conditions ?

l.257 (and Fig. 3) If PDH is the major contributor of NADPH waste for ButOH-0, ButOH-1 and tpcBuOH, what is the major contributor of NADPH consumed in the NADPH waste?

Also related to this point, maybe the authors can expand on why the NADPH waste is 0 in BUTYR (Fig.3 B/C) versus non-zero in BUTAL. After all, these two models are one reaction consuming ATP apart

l.355 constraining both lower and upper boundaries?

l.377 “After cycle detection,..” What is the cutoff value here to decide that this particular set of reactions is now actually a futile cycle?

l.403 “BuOH-1 appears to be the most balanced..” most balanced in term of what metric? Redox balance? It would seem that the model having no flux towards biomass instead is actually the most balance redox-wise

Related to the above point, have the authors tried to include an artificial NADPH and/or an ATP sink reaction in the model to see how imbalanced the redox state is? It would be interesting to see if these reactions would have a non-zero flux in the model making no biomass for instance. It could also be used as a metric to determine redox imbalance in each model.

l.427 Same as above artificial sink reaction could prove that. If you add an ATP sink reaction, how does the CBA analysis of NADPH vary and vice versa

l.501 and paragraph. I think this is an unfair comparison between 13C-MFA and FVA variability. First, knocking-out a single gene would not probably push the reactions of the network to their maximal range as this is the case in silico. Secondly, this is certainly not the case with FBA as it predicts a long-term flux distribution (after many generations). Whereas the 13C-MFA data obtained from Havekorn et al. will provide flux data that should instead be compared to MOMA (Segre et al.).

Fig.7 Great results to guide pathway engineering (on a pure stochiometric aspect). From this figure, it seems it would be possible to determine an “optimal” ATP/NADPH ratio for each product. I would also suggest to add a marker showing where each model stand in each sub-plot

l.607 Can the authors hypothesize on why no solutions exist for ATP produced >2 in Fig. 7 G & H. Since product is set as objective function, does this mean that no solution exist? It would be good to indicate that there is no flux solution (leave the area blank) rather than implying no flux to product.

Reviewer #2: The authors had a very interesting idea to explore cofactor balance and its relation to maximizing the production of compounds. The authors leveraged FBA, CBA, FBA, and MFA to interrogate the Ecoli core and 8 addition engineered models with different combinations of pathways that leads to the biosynthesis of butanol. Each model had a different demand for ATP and NAD(P)H and the authors looked at how those change in redox demand effected flux distribution and yield under aerobic and anaerobic conditions. I think the idea is interesting, however there are some concerns associated with the analysis.

1) The comparison between WT and engineered strains is not a fair comparison due to the difference in objective functions. Would the results change if all of the engineered trains were constrained to the growth rate of the WT, and then maximizing the production of butanol?

2) FVA not conducted across all engineered strains. Single FBA/CBA flux distributions were used to draw conclusions about how redox demand effects the over flux distribution of the network. However, many other solutions exist that yield the same objective value, the singular solution that the author presented and analyzed will change based on the algorithm and solver that was used to solve the linear programming problem. Although the yields may be correct, the CBA may yield an incomplete picture. How much did the cofactor contributions change if FVA was performed?

3) NADPH and NADH are analyzed interchangeably, however pathways such as pentose phosphate replenishes NADPH and can contribute to the cofactor analysis.

Reviewer #3: This paper proposes a procedure to analyse the impact of metabolic engineering on the balance of co-factors, and use it to evaluate alternative metabolic engineering designs.

The results are interesting and the paper is well written.

However, the several computational procedures could be presented with more mathematical details:

1) In Section 4.2, I do not undersand why max vobjective is set equal to vobjective,lb (in this way it appears as a constraint, rather than a true maximization). What is vobjective,lb?

2) Still in section 4.2, the models are unconstrained (except for the stoichiometry and irreversibility of course): what is the number of degrees of freedom of the different models?

3) the text often mentions the "excessive flexibility of FBA". I think that this statement sounds a bit too strong, and vague at the same time. Per se, FBA is an excellent tool, all depends on the degrees of freedom of the problems under considerations.

Additional information or measurements will help reduce the underdeterminacy of the problem (which is a better concept than the flexibility of a method).

4) Is the CBA protocol available to the community as a Python function?

5) In FVA, why 99% of the maximal value of the objective function, and not 100%?

6) In section 4.4, C-13 labelling experiments by Haverkorn et al. (which is reference 44 and not 46) are used to constrain the models. Afterwards, it is refered as the "Rijsewijk dataset" which is unclear.

Could the authors elaborate on the fact that this dataset is appropriate to create constraints in their study?

The use of the flux values to create min-max bounds seems to be a big assumption.

What are the remaining degrees of freedom after the implementation of these constraints?

7) Figures 3 and 5 would be easier to interpret if the same scales were used in the graphs.

Reviewer #4: General comments:

The authors presented analyses on the cofactor balance of different production pathways for butanol and its precursors. The authors also organized the analysis steps into a workflow. By examining the energy or redox wastage the authors claimed that those flux distributions are not biologically realistic and made efforts to block such wastage like ATP hydrolysis and redox sinks and futile cycles in particular ways to find solutions in which energy and redox either go into products or growth, to predict more realistic yields. The assumption is based on previous studies that have suggested that when growth is the only major sink of energy and redox surplus, more materials (carbon, nitrogen, etc.) will be consumed by growth too and result in an overall lower yield.

Specific comments:

Though the analysis does generate some useful insight and results, and based on an assumption that is valid to a certain extent. There are major concern from the reviewer’s point of view.

Major comments

Although many observations and computations are solid, how the authors present their arguments or assumptions/hypotheses seem biased and not objectively based. And it does not represent a proper way of interpreting computational results.

1.1. One example is to directly take a certain type of FBA solutions (pFBA here) and use the flux values for intracellular reactions as the predictor and then draw conclusions from that. For example in section 2.4, the authors just take the values predicted for ATPM returned by the particular pFBA solution and claimed FBA predicted ATP wastage. The underlying philosophy of constraint-based modeling is that when we have more constraints from biochemical principles, like mass balance and reaction directionality, we can narrow down the range of phenotypes. But we should be always aware that since the system is mostly way underdetermined, we should never simply take a certain flux value and draw conclusions from that. A better way is to look at the FVA range, which the authors actually did. And from Table S1, FBA actually did not preclude solutions with minimum ATPM only (min values in all cases are 7.6). This is actually consistent with what the authors found. They were able to fix ATPM value and still get max. yields the same as before. The authors can present the FVA ranges and then say that they include metabolic flux distributions that represent energy and redox wastage but should not simply take a particular solution from FBA and say it predicted wastage since no wastage does lie within the prediction of FBA. It just did not exclude wastage prediction.

1.2. Following the last comment, lines 192 – 194 claims ‘Five out of eight models surprisingly had the same theoretical maxima under aerobic conditions achieved through considerable adjustments to the network flux distributions’.

But the authors looked at only one solution using pFBA for each model. The possibilities that there are other solutions that are more similar to each other given the same yields cannot be ruled out. The authors should claim that only if they have tried to, for example, add the maximum yields as constraints and minimize the difference (in whether a reaction is used, or just sum of squared difference) between all models and still see that they are significantly different.

Also, from Table S2, only BuOH-0, BuOH-1 have the same max. of 10 mol/10 mol glucose, while tpcBuOH, BuOH-2 and fasBuOH have lower yield. Did the author include also CROT, BUTYR and BUTAL which produce different end products (though they are precursors of butanol)? But still they are not quite comparable especially they involve cofactors that directly impact the final yield.

2. Line 303 asks “Is bacterial metabolism that flexible”? The authors did not directly answer this but from the analyses afterward it is pretty apparent that their answer was ‘NO’. But the reviewer does not see very strong evidence about that. The reviewer hopes that this is a misunderstanding originating from how the authors interpret the flux distributions from FBA. From the text later, the reviewer could see that the authors argued that it is not possibly that flexible because there is energy and redox surplus that go into wastage sink or futile cycles which are not biologically realistic. The authors argue that they should go into growth. So growth must accompany the product formation. Well, that should be obvious enough for most metabolic engineers. No growth no nothing no matter the product generates energy or redox surplus. But mostly metabolic engineers look at theoretical yield because we hope that the pathway can be active under the condition of growth or after growth. So under growing conditions, is that possible for the bacteria to include very different pathways with different cofactor balance profiles to the product formation? The reviewer tends to say yes. Just that the absolute fluxes through the pathways could be different depending on how well it is coupled to growth. Even not under growth conditions, the reviewer does not see the reason why those pathways are necessarily not feasible. Since they are engineering pathways anyway, there could be engineering efforts to further improve that. For example there have been work to introduce ATPase activity to consume ATP in resting cells and observed enhanced fluxes.

3. The argument in Lines 336 – 342 for “FBA having more flexibility than what would be expected in reality” seems highly speculative. Is the ‘flexibility’ that the authors were referring to the same as the ‘flexibility’ that ref. 27 (Ghosh, PLOS ONE, 2011)? It seems that in that article restoring the ‘flexibility’ is something desirable for the cells while here the authors were saying such flexibility was not realistic. There seemed to be some misinterpretation without proper definition. By the way the reference for ref. 27 needs fixing.

4. Lines 345-346 are misleading. Since the authors only used the E. coli core model, which does not have glycerol in the model, of course the solutions cannot resort to glycerol production. And it is just very intuitive that FBA will simply not produce biomass to consume the energy or redox in excess. Biomass production competes cellular resource with the pathways for producing many compounds because in addition to energy or redox, biomass production involves the synthesis of many metabolites and producing any biomass will very probably reduce the maximum yield of butanol, contradicting to the assumption of optimal yield when the authors ran pFBA.

5. Section 2.7 comparing the ranges from FBA and MFA could be over-/mis-interpretation of data. Metabolic fluxes are almost never directly measured (not that the reviewer knows). Flux ratios between pathways are usually first measured (this point included the assumption of which pathways to include) and then fitted into a stoichiometric model (usually constructed manually to have a degree of freedom <= 0 after constrained by the measured flux ratios). Therefore the observation that MFA flux ranges are narrower than FBA ranges by direct comparison are not necessarily valid because the models used to interpret the data or do predictions are different, which impact the results a lot (or at least to an unknown extent). Though I think the authors’ conclusion of MFA narrowing down FBA ranges must be correct, the reason is simply because MFA flux ratios give additional information that an FBA model does not have by default. But comparing directly this way without any elaboration gives a biased understanding. Did the authors examine carefully the original stoichiometric model used for fitting the flux data and check that it still largely applies to the E. coli core model? Or just fit the data into the current model to check potential range?

Besides the interpretation of computation results, there are some major concerns in the computation methods used.

6.1. Related to the first point, the CBA algorithm only analyzed a particular solution from FBA. It is not a systematic analysis. There was a previous study introducing a technique called flux-sum analysis (Chung and Lee, BMC Syst. Biol., 2009, https://bmcsystbiol.biomedcentral.com/articles/10.1186/1752-0509-3-117) to address this issue. Basically one should find the ‘cofactor turnover’ range by minimizing and maximizing the sum of all cofactor score (as defined by the authors) to fully describe the flux through a cofactor. Similarly, for the energy and redox balance profile breakdown, one should minimizing and maximizing the sum of all cofactor score across reactions belong to a particular category and present a range. Simply comparing one particular flux distribution from each model is not representative.

6.2. Why did the authors define CO2-releasing NADH-producing reactions such as PHDc as waste? It is hard to understand. Looking at the right plot in Figure 3B, a large portion of the ‘NADH waste’ (+ve) does go into ‘NADH target’ (-ve). Is that a proper definition of waste?

7.1. The procedure to eliminate futile cycles described in lines 372 to 384 is problematic. Bounding the fluxes with the values from the wild-type flux distribution, which was obtained by maximizing biomass production, will probably make the predicted fluxes look like the growth metabolism. This also potentially excludes other reactions that can be used but not active under the condition of maximum biomass production.

7.2. There are well-established techniques to add constraints to exclude all futile cycles while not excluding any feasible flux distributions without futile cycles. For example, adding integer cuts iteratively instead of adding bounds on specific reactions. More systematically, one can use the loopless FBA formulation (first published in Schellenberger et al., Biophysics Journal, 2011 and a number of performance improved formulation later). The formulation usually blocks completely balanced cycles which has delta G = 0. For the purpose here, one can simply calculate the null space which excludes the cofactors of interest (ATP, ADP, Pi, H2O, H+, NADH and NAD+) and use the same formulation accordingly. Then all cycles whose net effect are consumption or production of the cofactors will be prohibited in the solution. See the Methods section in Ng et al., Scientific Report, 2019 (https://www.nature.com/articles/s41598-019-38836-9) for an example. Basically you just equations (6) - (10) in the method section

8. Why did the authors only look at the E. coli core model? As stated in comment 4, that potentially causes some systematic errors because a genome-scale E. coli model could probably predict energy or redox consumption through glycerol production but the current model just has no such reaction. Does the argument still hold in general?

9. Overall, the reviewer does think the authors had some useful ideas presented but the way of presentation and interpretation is somehow biased and the computational method is not systematic enough to draw proper conclusions. The reviewer suggests that the authors state clearly their assumptions in exact terms. The wording in the abstract and introduction is not clear. Not until in the middle of the results the reviewer could understand what the issues they were trying to address. Simply state that energy and redox surplus associated to a certain engineering pathway should be consumed only by growth but not artificial sinks in the model (ATPM, etc.) and futile cycles which are tightly regulated. So the authors developed a method to first check the RANGE of energy and redox balance profile (by doing the flux-sum analysis with reactions categorized into sets for growth, product, waste, etc.) to see if a pathway has energy or redox surplus. Then formulate an optimization method based on loopless FBA for predicting pathways and theoretical yields in the absence of unconstrained high wastage flux and futile cycles. For the MFA constraint added, the best would be to add the originally derived flux ratios as constraints. If not possible at least the assumptions or applicability should be carefully studied and discussed. The reviewer would appreciate in the more general and systematic effort to find out pathways that couple growth to energy and redox surplus to have a more realistic yield and pathway prediction.

Minor comments:

- Please include the gene/reaction names in Figure 1. It should not difficult to explicitly show and name all eight different routes with good use of colors. Or beside the main figure, add thumbnail versions of all different pathways without the detailed names but simply arrows for the particular routes

- Refer to Tables S2 and S3 for the FBA-calculated yields between line 185 and line 188.

- What does the % yields between lines 185 and 188 mean? They are neither weight nor carbon yield. If it is simply molar ratio of butanol to glucose, that should not be reported as a percentage. Like probably no one will say the theoretical yield of glycolysis from glucose to pyruvate is 200% just because 1 mol of glucose yields 2 mol of pyruvate. The reviewer suggests using carbon yield instead, i.e., 1 mol glucose to 1 mol butanol means 66.67% carbon/carbon yield. The same goes for Figure 3A.

- Line 279: please state the reaction name and ID corresponding to Tables S4 or S5.

- Line 291: which figure the authors were referring to? Figure 3B or 3C, ATP or NADH?

- Lines 293 and 295: it reads incorrect to the reviewer to call PFL and PDH the same/identical reaction. In particular for these two reactions. The analysis can be made more relevant by turning on PFL only in anaerobic conditions, which is a consensus on PFL activity.

- Section 2.3 is not well-oriented. Not until the very last sentence only the reviewer can get the message that the authors were trying to assess how practical/biologically realistic the solutions from pFBA are. Please include a topic sentence telling what the authors want to examine and what they did before going deep into the details. For example, saying the following in the beginning of that paragraph will orient the readers much better (a pretty arbitrary construction that does not seems right): ‘Since realistic bacterial metabolism usually exhibits only a limited flexibility [if that is the assumption the authors have], we examined the plasticity in the FBA solutions for different pathways to determine if they together fit into a realistic description of bacterial metabolism.’

- The text in lines 312 (“all models displayed a flux increase through the ATPM reaction …”) and 324 (“models tpcBuOH, BuOH-2 and fasBuOH did not show any energy surplus”) and the data in Table S2 (ATPM = 7.6 for WT, tpcBuOH, BuOH-2 and fasBuOH) seem to be contradictory.

- The math and the algorithm are not clearly presented in the Methods section (4.2) and not in Figure 8. For example, CFS = co-factor stoich. x flux should be written using the standard symbols which have already been introduced in the FBA section, for example:

o CFS_i,j = S_{i,j}v_j for cofactor i in reaction j.

o CFS_i = \\sum_{reactin j, if CFS_i,j > 0} CFS_i,j, or simply \\sum_{j} abs(CFS_i,j)/2 as in the flux-sum analysis paper defined (Chung and Lee, BMC Syst. Biol., 2009, https://bmcsystbiol.biomedcentral.com/articles/10.1186/1752-0509-3-117

**Have all data underlying the figures and results presented in the manuscript been provided?**

Reviewer #1: Yes

Reviewer #2: Yes

Reviewer #3: Yes

Reviewer #4: Yes

PLOS authors have the option to publish the peer review history of their article (what does this mean?). If published, this will include your full peer review and any attached files.

Reviewer #1: No

Reviewer #2: No

Reviewer #3: No

Reviewer #4: No
---

## [Decision Letter · Decision Letter 1]

6 Jul 2020

Dear Dr Jones,

We are pleased to inform you that your manuscript 'In silico  co-factor balance estimation using constraint-based modelling informs metabolic engineering in Escherichia coli' has been provisionally accepted for publication in PLOS Computational Biology.

Best regards,

Anders Wallqvist

Associate Editor

PLOS Computational Biology

Jason Papin

Editor-in-Chief

PLOS Computational Biology

Reviewer's Responses to Questions

**Comments to the Authors:**

Reviewer #1: I thank the authors for their detailed comments and revisions. It is my opinion that the article is improved by the additional data and revised text. There is now more clarity in describing motivation, methodology, and results. All of the technical questions have been addressed.

Reviewer #2: Very interesting work, great job!

**Have all data underlying the figures and results presented in the manuscript been provided?**

Reviewer #1: Yes

Reviewer #2: Yes

PLOS authors have the option to publish the peer review history of their article (what does this mean?). If published, this will include your full peer review and any attached files.

Reviewer #1: No

Reviewer #2: No

---

## [Editor Report · Acceptance letter]

3 Aug 2020

PCOMPBIOL-D-19-02145R1 

In silico  co-factor balance estimation using constraint-based modelling informs metabolic engineering in Escherichia coli

Dear Dr Jones,

I am pleased to inform you that your manuscript has been formally accepted for publication in PLOS Computational Biology. Your manuscript is now with our production department and you will be notified of the publication date in due course.

With kind regards,

Matt Lyles
